# Interaction-aware Representation Modeling with Co-occurrence Consistency for Egocentric Hand-Object Parsing

**Yuejiao Su, Yi Wang[*], Lei Yao, Yawen Cui, Lap-Pui Chau[*]**
The Hong Kong Polytechnic University
{yuejiao.su, rayyoh.yao}@connect.polyu.hk
{yi-eie.wang, yawen.cui, lap-pui.chau}@polyu.edu.hk

## Abstract

A fine-grained understanding of egocentric human-environment interactions is crucial for developing next-generation embodied agents. One fundamental challenge in this area involves accurately parsing hands and active objects. While transformer-based architectures have demonstrated considerable potential for such tasks, several key limitations remain unaddressed: 1) existing query initialization mechanisms rely primarily on semantic cues or learnable parameters, demonstrating limited adaptability to changing active objects across varying input scenes; 2) previous transformer-based methods utilize pixel-level semantic features to iteratively refine queries during mask generation, which may introduce interaction-irrelevant content into the final embeddings; and 3) prevailing models are susceptible to "interaction illusion", producing physically inconsistent predictions. To address these issues, we propose an end-to-end Interaction-aware Transformer (InterFormer), which integrates three key components, *i.e.*, a Dynamic Query Generator (DQG), a Dual-context Feature Selector (DFS), and the Conditional Co-occurrence (CoCo) loss. The DQG explicitly grounds query initialization in the spatial dynamics of hand-object contact, enabling targeted generation of interaction-aware queries for hands and various active objects. The DFS fuses coarse interactive cues with semantic features, thereby suppressing interaction-irrelevant noise and emphasizing the learning of interactive relationships. The CoCo loss incorporates hand-object relationship constraints to enhance physical consistency in prediction. Our model achieves state-of-the-art performance on both the Ego-HOS and the challenging out-of-distribution mini-HOI4D datasets, demonstrating its effectiveness and strong generalization ability. Code and models are publicly available at `https://github.com/yuggiehk/InterFormer`.

## 1 Introduction

Recent advances in personal terminal devices such as GoPros and head-mounted devices (HMDs) have driven a significant increase in sharing first-person view (FPV, or egocentric) images and videos on various social media platforms (Xu et al., 2024a; Fan et al., 2025; Chen et al., 2025; Cartillier et al., 2021). In response, the research community has released large-scale FPV datasets including Ego4D (Grauman et al., 2022), EPIC-KITCHENS (Damen et al., 2018), and HOI4D (Liu et al., 2022b). Unlike third-person view (TPV) or exocentric perspective data (Kim et al., 2025; Lei et al., 2024), egocentric content directly captures the immersive visual information experienced by the camera wearer, providing details about their interactions with the surrounding environment (Huang et al., 2024; Shi et al., 2024). To better understand human behavior from the egocentric perspective, a fundamental challenge lies in accurately parsing the hands and objects involved in interaction, which is the main objective of the Egocentric Hand-Object Segmentation (EgoHOS) task (Zhang et al., 2022; Su et al., 2025a). By focusing on pixel-level segmentation of hands and interacting objects, this fine-grained analysis is able to interpret the complex dynamics of human-environment engagement, forming a foundational capability for next-generation technologies such as assistive

---

[*]Corresponding author.

agents (Yang et al., 2025; Zhou et al., 2025), embodied AI (Dang et al., 2025), and AR/VR systems (Zhao et al., 2024).

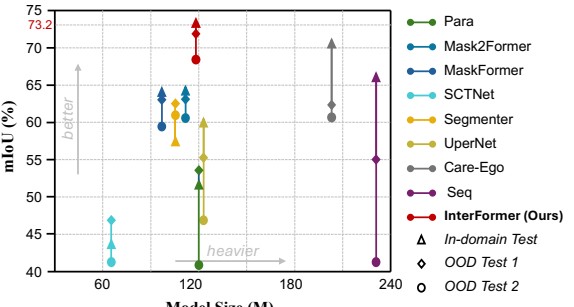

Figure 1: **Model size vs. mIoU for InterFormer compared to other methods.** Evaluations use EgoHOS indomain (In-domain), EgoHOS out-of-domain (OOD Test 1), and mini-HOI4D (OOD Test 2) datasets.

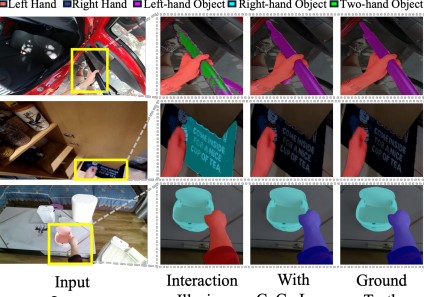

Figure 2: **Illustration of interaction illusion**, in which segmentation results violate real-world causal dependencies between hands and objects.

Existing approaches for parsing hands and interacting objects can be broadly categorized into convolution-based, transformer-based, and multi-modal large language model (MLLM)-based methods. Convolution-based models (Zhao et al., 2025; Xu et al., 2024b) are proposed in earlier stages, exhibiting limited performance in handling the complexities of the egocentric vision due to their inherent constraints in capturing long-range dependencies. With the recent advancement of Multimodal Large Language Models (MLLMs), MLLM-based methods (Su et al., 2025b) have gained attention for their powerful representational capacity. However, MLLM-based models typically incur substantial computational overhead and parameter costs. In contrast, transformer-based approaches (Zhang et al., 2022; Su et al., 2025a; Leonardi et al., 2024; Cheng et al., 2022a; Xie et al., 2021) offer a more favorable trade-off between accuracy and model complexity, effectively capturing long-range interactions while maintaining manageable parameter efficiency.

Despite the promising progress of transformer-based approaches, several key limitations remain unsolved. **First**, a primary issue lies in query initialization, which plays an essential role in transformer-based frameworks such as DETR (Carion et al., 2020; Zhu et al., 2021). Current methods typically initialize queries using either sampled image features (Zhou et al., 2022) or learnable parameters (Cheng et al., 2022a; Shah et al., 2024). The former often introduces irrelevant background information, while the latter provides a set of stable but static queries after sufficient training. Consequently, both approaches exhibit limited adaptability to dynamically changing active objects across diverse input scenes. **Second**, current methods predominantly rely on dense pixel-level semantic features from the input image, implicitly extracting target information through attention operations in the transformer decoder for mask prediction (Cheng et al., 2022a; Su et al., 2025a). However, such generic semantic features are fundamentally limited to answering "*what is it*" rather than *whether it is in interaction*. This semantic bias inevitably introduces substantial interaction-irrelevant noise, ultimately degrading segmentation accuracy. **Third**, existing methods (Zhang et al., 2022; Cheng et al., 2022a; Su et al., 2025a) often exhibit a logical defect termed as the interaction illusion. As illustrated in Figure 2, an object is predicted as manipulated by both hands even when the right hand is not detected. Such outcomes are inconsistent with real-world physical plausibility, leading to a significant drop in performance.

To address the aforementioned limitations, we propose the **Inter**action-aware Trans**Former** (**InterFormer**), a novel end-to-end framework for parsing hands and interacting objects from an egocentric perspective. In contrast to prior methods that rely primarily on semantic features (Su et al., 2025a; Cheng et al., 2022a; Zhang et al., 2022), we first introduce the Interaction Prior Predictor (IPP), an auxiliary branch trained to estimate interaction boundaries. This branch extracts preliminary boundary-guided features that coarsely localize hand-object contact regions and capture initial interaction characteristics. However, these rough boundary-guided features alone are insufficient for accurately distinguishing hands and their interacting objects. Therefore, we further propose a **D**ynamic **Q**uery **G**enerator (**DQG**) and a **D**ual-context **F**eature **S**ynthesizer (**DFS**) to shift the

model's focus to distinguishing interaction representation learning. Specifically, the DQG grounds query initialization within the dynamic spatial cues of ongoing interactions. This module selects semantic embeddings that demonstrate strong similarity with boundary-guided features and integrates them with learnable parameters, producing intrinsically interaction-aware queries that enable flexible adaptation to diverse hands and interactive objects across varying scenes. To address the noise caused by relying solely on pixel-level semantic features for query refinement, we propose the DFS. This module synthesizes coarse interaction boundary cues with semantic features, effectively suppressing interaction-irrelevant information and refocusing the model on essential contact relationships. Furthermore, to mitigate the interaction illusion problem, we design a **Co**nditional **Co**-occurrence (**CoCo**) loss that incorporates hand-object contact constraints to ensure physically plausible and accurate segmentation. We conduct extensive experiments to evaluate the effectiveness of our method. The results show that our InterFormer consistently surpasses all competing approaches across all evaluation metrics, achieving relative improvements of 2.42%, 5.09%, and 11.4% on the EgoHOS in-domain test set (Zhang et al., 2022), the EgoHOS out-of-domain test set (Zhang et al., 2022), and the out-of-distribution (OOD) mini-HOI4D dataset (Su et al., 2025a), respectively. These findings demonstrate the superior performance and robust generalization capability of our approach, as illustrated in Fig. 1. The main contributions of our InterFormer can be concluded as:

- We establish a novel query initialization paradigm, DQG, which generates intrinsically interaction-aware queries by fusing coarse interaction-aligned semantic embeddings with learnable parameters, enabling dynamic adaptation to hands and diverse active objects across varying scenes.

- The proposed DFS introduces an interaction-centric refinement mechanism that purifies semantic embeddings through boundary-guided feature fusion, effectively suppressing interaction-irrelevant noise and refocusing the model on contact relationships.

- We introduce a novel CoCo loss that encodes intuitive hand-object contact constraints into the learning process. By penalizing physically implausible co-occurrences, the CoCo loss significantly mitigates the "interaction illusion" problem and improves segmentation consistency.

- Extensive evaluations on EgoHOS and mini-HOI4D benchmarks confirm that InterFormer achieves state-of-the-art (SOTA) performance and exhibits strong generalization across in-domain and out-of-distribution settings.

## 2 RELATED WORK

Egocentric images and videos Narasimhaswamy et al. (2024) provide a unique perspective on human-environment interactions, capturing how individuals manipulate objects with their hands in natural, unstructured, real-world settings (Plizzari et al., 2025; Dang et al., 2025). The study of egocentric vision is essential for developing advanced intelligent embodied agents and has attracted growing interest from academic and industrial research communities. In response, several large-scale egocentric datasets have been introduced to support data-driven modeling of human behavior, such as Ego4D (Grauman et al., 2022), EPIC-KITCHENS (Damen et al., 2018), and HOI4D (Liu et al., 2022b). These datasets provide foundational resources for a wide range of tasks, including action recognition (Peirone et al., 2025), video captioning (Ohkawa et al., 2025), action anticipation (Lai et al., 2024b), affordance learning (Luo et al., 2024), and hand-object interaction interpretation (Leonardi et al., 2024). Despite the availability of these new datasets, the volume of FPV data remains considerably smaller than that of TPV datasets, limiting the training of deep models. To mitigate this data scarcity, several studies have explored cross-view representation learning, aiming to transfer view-invariant features from the TPV domain to the FPV domain (Jia et al., 2024; Liu et al., 2024; Li et al., 2024). However, such approaches typically rely on precisely aligned multi-view recordings, which are challenging to collect at scale due to hardware and synchronization constraints. Concurrently, some methods have integrated complementary multi-modal signals to enrich egocentric representation (Wang et al., 2025; Ramazanova et al., 2025), e.g., gaze (Lai et al., 2024a), audio (Jia et al., 2024), and textual descriptions (Hong et al., 2025; Wang et al., 2025). These modalities provide auxiliary contextual cues that can compensate for visual ambiguities in FPV data.

Recent advances in transformer architectures have significantly promoted the research in egocentric hand-object interaction (EgoHOI) by enabling more effective modeling of long-range dependencies

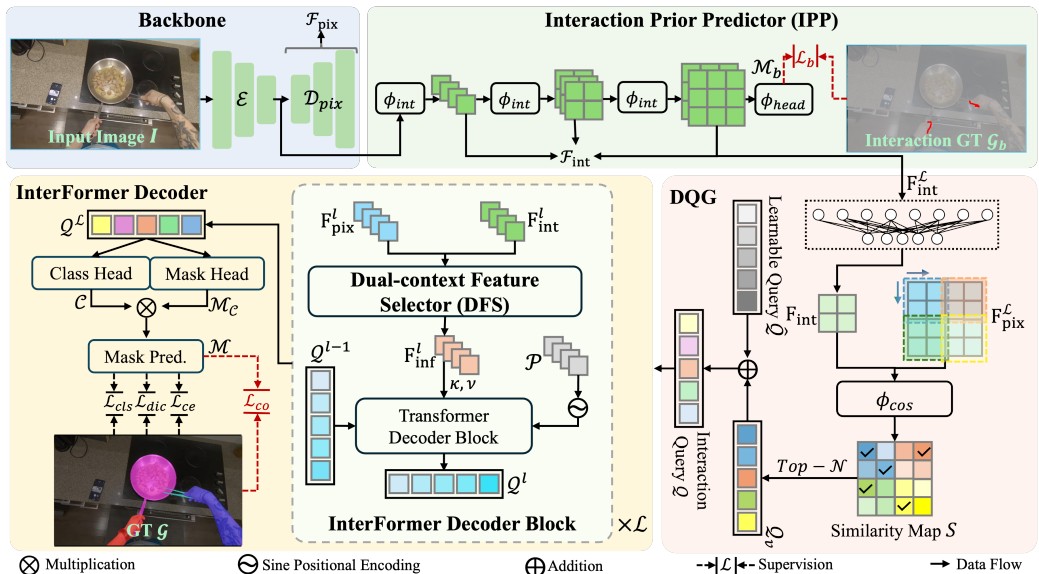

Figure 3: **Architecture of our end-to-end InterFormer.** Given an input egocentric image, a backbone network first extracts global and multi-scale pixel-level features. We add an additional IPP branch to extract coarse boundary-guided representations that characterize the interaction. Subsequently, the DQG produces robust and dynamic queries by integrating interaction-relevant contextual information with learnable parameters. Finally, these queries and extracted features are fed into the InterFormer decoder, which employs the DFS to refine interaction-aware representations and generate the final segmentation masks. The overall end-to-end architecture is supervised by the classification loss $\mathcal{L}_{cls}$, dice loss $\mathcal{L}_{dic}$, cross entropy loss $\mathcal{L}_{ce}$, IPP loss $\mathcal{L}_b$, and CoCo loss $\mathcal{L}_{co}$.

and complex visual relationships (Lai et al., 2024a; Roy et al., 2024; Cao et al., 2024). Despite these improvements, a fundamental limitation remains: most existing methods lack explicit and structured mechanisms for capturing the interactive relationships between hands and objects Cheng et al. (2022a); Su et al. (2025a); Zhang et al. (2022). As a result, they often produce inaccurate or physically implausible predictions.

## 3 METHODOLOGY

The EgoHOS task aims to parse the left hand $\mathcal{M}_{lh}$, right hand $\mathcal{M}_{rh}$, and objects in contact within an egocentric image $\mathcal{I} \in \mathbb{R}^{H \times W \times 3}$. The categories of interacting objects include left-hand objects $\mathcal{M}_{lo}$ (objects that interact only with the left hand), right-hand objects $\mathcal{M}_{ro}$ (objects that interact only with the right hand), and two-hand objects $\mathcal{M}_{to}$ (objects that are contacted by both hands).

### 3.1 OVERVIEW

This paper presents InterFormer, a novel approach for precisely parsing hands and interacting objects in egocentric images. The overall architecture is shown in Figure 3. Given an egocentric image, the backbone first extracts global and multi-scale pixel-level features. Unlike existing approaches that rely solely on semantic features, we introduce an additional interaction prior predictor, which is explicitly supervised by the interaction boundary ground truth $\mathcal{G}_b$ to guide the network to concentrate on hand-object contact regions and model boundary-guided cues. However, these rough boundary-guided features alone are insufficient for accurately distinguishing hands and their interacting objects. Therefore, we further design a dynamic query generator and a dual-context feature selector to explicitly refine and enrich the interaction representation. Specifically, the DQG grounds query initialization within the dynamic spatial cues of ongoing interactions, generating robust and dynamic queries for hands and manipulated objects. The generated queries along with the extracted features are transmitted into the InterFormer decoder, which integrates the DFS to refine interaction-aware representations and produce the final segmentation masks. Details of each component are provided in the following subsections.

**Backbone.** The backbone of InterFormer extracts global and pixel-level features from an input egocentric image $\mathcal{I}$. Specifically, we use a Swin (Liu et al., 2021) Transformer encoder $\mathcal{E}(\mathcal{I}; \theta_e)$ to obtain global features $\mathbf{F}_g \in \mathbb{R}^{H_g \times W_g \times C_g}$, where $\theta_e$ is the parameter. Next, a deformable DETR transformer (Zhu et al., 2021) serves as the pixel decoder $\mathcal{D}_{pix}$, generating multi-scale pixel-level features $\mathcal{F}_{pix} = \left\{ \mathbf{F}_{pix}^l \in \mathbb{R}^{H_p^l \times W_p^l \times C_p^l} \mid l \in \{1, 2, \cdots, \mathcal{L}\} \right\}$, where $H_p^l$, $W_p^l$, and $C_p^l$ denote the height, width, and channel dimensions at each scale.

**Interaction Prior Predictor.** After extracting pixel-level features, most existing methods send them directly into the transformer decoder for prediction Cheng et al. (2022a); Su et al. (2025a); Zhang et al. (2022). However, since different actions involve different interacting objects, identifying active objects cannot rely solely on semantic information, but must depend on their relationship with the hand. Therefore, we introduce an interaction prior predictor branch to roughly localize the hand-object interaction. This branch takes the global feature $\mathbf{F}_g$ as input and predicts the interaction boundary (Zhang et al., 2022), *i.e.*, the overlapping region between hands and interacting objects, using a cascaded U-Net-style decoder $\phi_{int}$ followed by stacked convolutional layers $\phi_{head}$. The output is an interaction boundary map $\mathcal{M}_b$ supervised using binary cross-entropy loss: $\mathcal{L}_b = \mathcal{L}_{bce}(\mathcal{M}_b, \mathcal{G}_b)$, where $\mathcal{G}_b$ denotes the ground truth of the interaction boundary generated by the intersection of dilated hand and object masks. By predicting interaction boundaries, the model achieves coarse spatial localization of hand-object interaction regions, thereby generating preliminary boundary-guided features $\mathcal{F}_{int} = \left\{ \mathbf{F}_{int}^l \in \mathbb{R}^{H_b^l \times W_b^l \times C_b^l} \mid l \in \{1, 2, ..., \mathcal{L}\} \right\}$, which provide essential spatial constraints for subsequent interaction modeling. However, these features are insufficient for accurately distinguishing hands and their interacting objects. Therefore, we further design a dynamic query generator and a dual-context feature selector to explicitly refine and enrich the interaction representation.

## 3.2 DYNAMIC QUERY GENERATOR

Query initialization is crucial in transformer-based methods, as it dominates the model's attention to the most relevant information and significantly influences the learning procedure. Some approaches (Cheng et al., 2022a; Shah et al., 2024; Li et al., 2023; Jain et al., 2023; Zhang et al., 2023; 2021) employ learnable parameters as queries, offering robustness and stability but often leading to slower convergence due to delayed feature alignment. In contrast, others use sampled features (Zhou et al., 2022; Cheng et al., 2022b;c; Fu et al., 2024) to initialize queries, which offers adaptability to input content but potentially introduces noise from irrelevant or ambiguous regions. More importantly, neither method explicitly encodes hand-object interactive relationships in queries, resulting in a lack of adaptation to hands and diverse interacting objects in varying input images.

To address these limitations, we propose the Dynamic Query Generator (DQG). The key innovation of this module lies in grounding query initialization in the dynamic spatial cues of interactions through a two-stage process. First, it extracts interaction-relevant content by selecting semantic embeddings that demonstrate strong correspondence with boundary-guided features, ensuring the selection captures genuine contact relationships rather than relying solely on semantic information as in traditional feature-sampling methods. Second, it synthesizes these selected features with learnable parameters to generate the final interaction-aware queries. Specifically, the last-layer pixel-level feature $\mathbf{F}_{pix}^{\mathcal{L}} \in \mathbb{R}^{H_p^{\mathcal{L}} \times W_p^{\mathcal{L}} \times C_p^{\mathcal{L}}}$ and boundary-guided feature $\mathbf{F}_{int}^{\mathcal{L}} \in \mathbb{R}^{H_b^{\mathcal{L}} \times W_b^{\mathcal{L}} \times C_b^{\mathcal{L}}}$ are first aligned in the channel dimension via a multi-layer perceptron (MLP), resulting in $\mathbf{F}_{int} \in \mathbb{R}^{\frac{H_p^{\mathcal{L}}}{n} \times \frac{W_p^{\mathcal{L}}}{n} \times C_p^{\mathcal{L}}}$. Consequently, we uniformly partition $\mathbf{F}_{pix}^{\mathcal{L}}$ into $n \times n$ non-overlapping sub-regions along the height and width dimensions and compute the cosine similarity between each sub-region and $\mathbf{F}_{int}$. This procedure produces a dense similarity map $S \in \mathbb{R}^{H_p^{\mathcal{L}} \times W_p^{\mathcal{L}}}$ defined as follows:

$$S = \frac{\langle \mathbf{F}_{int}, \mathbf{F}_{pix}^{\mathcal{L}}(i,j) \rangle}{\|\mathbf{F}_{int}\| \cdot \|\mathbf{F}_{pix}^{\mathcal{L}}(i,j)\|}, i,j \in \{1, 2, \ldots, n\}, \tag{1}$$

where $\mathbf{F}_{pix}^{\mathcal{L}}(i,j)$ denotes the feature vector at the $(i,j)-th$ sub-region. Next, we select the $\mathcal{N}$ highest similarity values in $S$ and extract the corresponding feature vectors from $\mathbf{F}_{pix}^{\mathcal{L}}$ to form a query vector $\mathcal{Q}_v \in \mathbb{R}^{\mathcal{N} \times C_p^{\mathcal{L}}}$, which encodes salient interactive regions. This intermediate query is

then combined with a learnable parameter vector through element-wise addition to produce the final interaction query $\mathcal{Q} \in \mathbb{R}^{\mathcal{N} \times C_P^{\mathcal{L}}}$, which serves as the initial input to the transformer decoder.

The proposed DQG extracts interaction-relevant content by measuring the similarity between image features and boundary-guided representations. The extracted features, which correspond to hand-object interaction regions, are combined with learnable parameters to construct robust and adaptive queries. This approach enables the model to explicitly generate queries based on dynamic interaction contexts rather than static object categories. Consequently, for input images with varying active objects, queries can be constructed according to the hand-object interaction regions. The inclusion of learnable parameters further enhances the flexibility of the query formulation process.

## 3.3 DUAL-CONTEXT FEATURE SELECTOR

After the query initialization process, most current methods rely on dense pixel-level semantic features to implicitly learn target information through attention operations in the transformer decoder for mask prediction. However, such generic semantic features are fundamentally limited to answering "*what is it*" rather than *whether it is in interaction*. This semantic bias inevitably introduces substantial interaction-irrelevant noise, ultimately degrading segmentation accuracy.

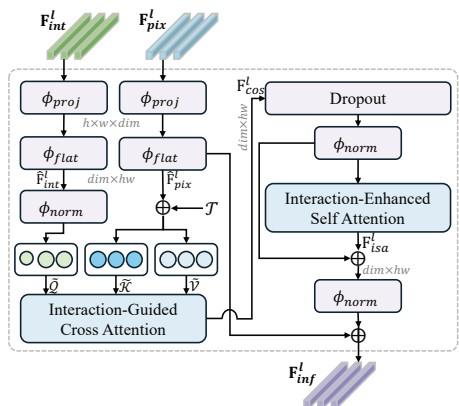

To address this limitation, we introduce a dual-context feature selector within each InterFormer decoder layer to explicitly enhance interaction understanding. As depicted in Fig. 4, for the l-th layer, the DFS inputs the pixel-level feature $\mathbf{F}_{pix}^l \in \mathbb{R}^{H_P^l \times W_P^l \times C_P^l}$ and the corresponding preliminary boundary-guided feature $\mathbf{F}_{int}^l \in \mathbb{R}^{H_b^l \times W_b^l \times C_b^l}$. Both features are first projected to the same dimension and reshaped to size $h \times w \times dim$. Next, they are flattened along the spatial dimensions $h$ and $w$, yielding $\hat{\mathbf{F}}_{int}^l$ and $\hat{\mathbf{F}}_{pix}^l$. A learnable positional parameter $\mathcal{T} \in \mathbb{R}^{(hw) \times dim}$ is then added to $\hat{\mathbf{F}}_{pix}^l$ to improve robustness. Subsequently, an interaction-guided cross-attention mechanism is deployed to fuse semantic and interactive information, where the query $\tilde{\mathcal{Q}}$ is derived from the boundary-guided interaction feature, while the key $\tilde{\mathcal{K}}$ and value $\tilde{\mathcal{V}}$ are computed from the pixel-level feature. Specifically:

Figure 4: Detailed architecture of Dual-context Feature Selector (DFS).

$$\hat{\mathbf{F}}_{pix}^l = \phi_{flat}(\phi_{proj}(\mathbf{F}_{pix}^l)), \tilde{\mathcal{K}}, \tilde{\mathcal{V}} = \phi_{cov}(\mathcal{T} + \hat{\mathbf{F}}_{pix}^l), \tag{2}$$

$$\tilde{\mathcal{Q}} = \phi_{cov}(\phi_{norm}(\phi_{flat}(\phi_{proj}(\mathbf{F}_{int}^l)))), \tag{3}$$

where $\phi_{flat}, \phi_{proj}, \phi_{cov}$, and $\phi_{norm}$ denote the flatten, linear projection, convolution, and normalization operations, respectively. Then, the interaction-guided cross-attention operation is performed using $\tilde{\mathcal{Q}}, \tilde{\mathcal{K}}$ and $\tilde{\mathcal{V}}$:

$$\mathbf{F}_{cos}^l = softmax(\frac{\tilde{\mathcal{Q}}\tilde{\mathcal{K}}^T}{\sqrt{dim}})\tilde{\mathcal{V}}. \tag{4}$$

Subsequently, the fused feature $\mathbf{F}_{cos}^l$ is passed through a dropout layer $\phi_{drop}$ and a normalization layer, followed by an interaction-enhanced self-attention module $\phi_{sa}(\cdot)$. This operation is identical to standard self-attention operation, which refines the feature representation by modeling long-range dependencies within the interaction-aware context, yielding a more discriminative output $\mathbf{F}_{isa}^l$. The final integrated feature $\mathbf{F}_{inf}^l$ is then computed through residual connection and normalization:

$$\mathbf{F}_{isa}^l = \phi_{sa}(\phi_{norm}(\phi_{drop}(\mathbf{F}_{cos}^l))), \tag{5}$$

$$\mathbf{F}_{inf}^l = \hat{\mathbf{F}}_{pix}^l + \phi_{norm}(\mathbf{F}_{isa}^l + \phi_{norm}(\phi_{drop}(\mathbf{F}_{cos}^l))). \tag{6}$$

By leveraging both interaction-guided cross-attention and interaction-enhanced self-attention, the DFS effectively fuses semantic content with structural interaction cues, thereby suppressing interaction-irrelevant information and learning more representative and interaction-aware features. In

each InterFormer decoder layer, the DFS-produced feature $\mathbf{F}_{inf}^{l}$ is used as the key and value within the transformer decoder block, while the query $\mathcal{Q}^{l-1}$ from the previous layer serves as the input query. This hierarchical attention mechanism enables progressive refinement of target localization through iterative feature alignment and interaction modeling. After $\mathcal{L}$ decoder layers, the final set of queries $\mathcal{Q}^{L}$ encodes rich target-specific representations, which are independently decoded into class predictions and mask reconstructions. Specifically, the predicted category distribution $\mathcal{C}$ and the class-agnostic mask $\mathcal{M}_{\mathcal{C}}$ are generated from $\mathcal{Q}^{L}$. The final segmentation output $\mathcal{M}$ is obtained by multiplication: $\mathcal{M} = \mathcal{C} \otimes \mathcal{M}_{\mathcal{C}}$, where $\otimes$ denotes channel-wise multiplication between the class logits and the corresponding masks.

### 3.4 Conditional Co-Occurrence Loss

**Interaction Illusion.** In real-world scenarios, the presence of a hand is a fundamental prerequisite for any hand–object interaction. For example, an object manipulated by the left hand can only be involved in interaction if the left hand itself is present and detected. However, existing methods (Zhang et al., 2022; Cheng et al., 2022a; Su et al., 2025a) often suffer from a phenomenon termed as interaction illusion, in which predicted interactions violate causal dependencies between hands and objects. As shown in the first row of Fig. 2, when the right hand is missing in the prediction, current models may incorrectly classify the interacting object as being operated by both hands, despite the absence of one hand. Such errors contradict basic physical constraints and undermine the reliability of segmentation systems in embodied AI applications.

To address this issue, we propose the Conditional Co-occurrence (CoCo) Loss, a novel supervision mechanism that explicitly enforces physically plausible hand–object segmentation by conditioning object predictions on the presence of the corresponding hand. Unlike probability-based penalties, our CoCo loss operates directly on the spatial extent of the predictions, i.e., the number of pixels in the predicted hand and object masks. We opt for this design based on the observation that the "interaction illusion" is fundamentally a macro-level logical error, which is more directly and effectively measured by the physical presence or absence (i.e., the pixel count) of the mask, rather than the average classification confidence across pixels. Guided by this principle, our CoCo loss is as follows: if the predicted mask for a given hand contains fewer pixels than a predefined threshold $\tau$ (indicating the absence of that hand), the loss penalizes any prediction of objects associated with that hand. This discourages implausible co-occurrence patterns, such as recognizing an object as the *left-hand object* when the left hand is not detected. Conversely, when the hand is confidently present (*i.e.*, pixel count exceeds $\tau$), the penalty is deactivated, allowing legitimate interactions to be learned without interference. This dynamic constraint guides the model toward causally consistent hand–object associations. The CoCo loss for the left and right hands is defined as:

$$\mathcal{L}_{co}^{left} = (1 - \mathbb{I}_{\{\mathcal{N}_{lh} > \tau\}}) \cdot (\mathcal{N}_{lo} - \mathbb{I}_{\{\mathcal{N}_{lh} > \tau\}} \cdot \mathcal{N}_{lo}) = (1 - \mathbb{I}_{\{\mathcal{N}_{lh} > \tau\}}) \cdot \mathcal{N}_{lo}, \quad (7)$$

$$\mathcal{L}_{co}^{right} = (1 - \mathbb{I}_{\{\mathcal{N}_{rh} > \tau\}}) \cdot (\mathcal{N}_{ro} - \mathbb{I}_{\{\mathcal{N}_{rh} > \tau\}} \cdot \mathcal{N}_{ro}) = (1 - \mathbb{I}_{\{\mathcal{N}_{rh} > \tau\}}) \cdot \mathcal{N}_{ro}, \quad (8)$$

where $\mathcal{N}_{lh}$, $\mathcal{N}_{lo}$, $\mathcal{N}_{rh}$, and $\mathcal{N}_{ro}$ denote the number of predicted pixels for the left hand, left-hand interacting object, right hand, and right-hand interacting object, respectively. $\mathbb{I}_{\{x\}}$ is the indicator function, which returns 1 if the condition is satisfied and 0 otherwise. Equations 7-8 illustrate the core mechanism of the CoCo loss, *i.e.,* the hand-first principle. To further extend this regulation to objects manipulated by both hands, we define the CoCo loss for two-hand interactions:

$$\mathcal{L}_{co}^{two} = (1 - \mathbb{I}_{\{N_{rh} > \tau \wedge N_{lh} > \tau\}}) \cdot (N_{to} - \mathbb{I}_{\{N_{rh} > \tau \wedge N_{lh} > \tau\}} \cdot N_{to}) = (1 - \mathbb{I}_{\{\mathcal{N}_{rh} > \tau \wedge N_{lh} > \tau\}}) \cdot \mathcal{N}_{to}, \quad (9)$$

where $\mathcal{N}_{to}$ denotes the number of predicted pixels for the object involved in two-hands manipulation, and the indicator function $\mathbb{I}_{\{N_{rh} > \tau \wedge N_{lh} > \tau\}}$ equals to 1 only when both hands are present. Thus, $\mathcal{L}_{co}^{two}$ penalizes predictions of two-hand objects unless both hands are simultaneously detected, enforcing a physically grounded co-activation prior. The proposed CoCo loss incorporates real-world physical constraints into the learning process, guiding the model toward logically consistent and physically plausible hand–object relationships.

**Overall Training.** The InterFormer framework is trained in a fully end-to-end manner, which is jointly supervised by the interaction boundary loss $\mathcal{L}_{b}$ and the proposed CoCo loss $\mathcal{L}_{co}$. Following established transformer-based segmentation approaches (Cheng et al., 2022a; Zhang et al., 2022),

Table 1: Comparison with SOTA methods on the EgoHOS in-domain test set measured by IoU ↑.

| Method | Type | Left Hand | Right Hand | Left-hand Object | Right-hand Object | Two-hand Object | Overall |
|---|---|---|---|---|---|---|---|
| Segformer (Xie et al., 2021) | T | 62.49 | 64.77 | 4.03 | 3.01 | 5.13 | $27.89_{(+45.33)}$ |
| SCTNet (Xu et al., 2024b) | T | 81.94 | 82.12 | 17.77 | 16.60 | 21.74 | $44.03_{(+29.19)}$ |
| Para (Zhang et al., 2022) | T | 69.08 | 73.50 | 48.67 | 36.21 | 37.46 | $52.98_{(+20.24)}$ |
| Segmenter (Strudel et al., 2021) | T | 82.20 | 83.28 | 46.22 | 34.79 | 51.10 | $59.52_{(+13.70)}$ |
| UperNet (Xiao et al., 2018) | C | 89.88 | 91.39 | 36.22 | 40.55 | 45.54 | $60.71_{(+12.51)}$ |
| Multi-UNet (Zhao et al., 2025) | C | 86.35 | 87.64 | 44.80 | 45.29 | 46.72 | $62.16_{(+11.06)}$ |
| MaskFormer (Cheng et al., 2022a) | T | 90.45 | 91.95 | 43.51 | 41.04 | 54.65 | $64.32_{(+8.90)}$ |
| OneFormer(Zhang et al., 2022) | T | 90.38 | 91.95 | 43.88 | 44.37 | 52.64 | $64.64_{(+8.58)}$ |
| Mask2Former (Cheng et al., 2022a) | T | 90.74 | 92.25 | 44.22 | 46.05 | 51.13 | $64.88_{(+8.34)}$ |
| Seq (Zhang et al., 2022) | T | 87.70 | 88.79 | **62.20** | 44.40 | 52.77 | $67.17_{(+6.05)}$ |
| ANNEXE (Su et al., 2025b) | L | 91.50 | 92.73 | 58.94 | **57.32** | 56.41 | $71.38_{(+1.84)}$ |
| Care-Ego (Su et al., 2025a) | T | 92.34 | **93.64** | 60.07 | 56.69 | 54.73 | $\underline{71.49}_{(+1.73)}$ |
| **InterFormer (Ours)** | T | **92.51** | 93.50 | 60.86 | 55.04 | **64.17** | **73.22** |

Table 2: Comparison results on the EgoHOS out-of-domain test set measured by IoU ↑.

| Method | Type | Left Hand | Right Hand | Left-hand Object | Right-hand Object | Two-hand Object | Overall |
|---|---|---|---|---|---|---|---|
| Segformer(Xie et al., 2021) | T | 71.97 | 71.44 | 7.60 | 5.00 | 4.91 | $32.18_{(+40.64)}$ |
| SCTNet(Xu et al., 2024b) | T | 87.12 | 86.29 | 31.18 | 19.70 | 13.32 | $47.52_{(+25.30)}$ |
| UperNet(Xiao et al., 2018) | C | 93.17 | 93.96 | 42.53 | 28.88 | 24.35 | $56.58_{(+16.24)}$ |
| Multi-UNet (Zhao et al., 2025) | C | 92.76 | 83.08 | 44.31 | 39.07 | 37.15 | $59.27_{(+13.55)}$ |
| Maskformer(Cheng et al., 2022a) | T | 92.69 | 94.02 | 51.81 | 39.84 | 39.43 | $63.56_{(+9.26)}$ |
| Mask2former(Cheng et al., 2022a) | T | 91.46 | 93.04 | 53.41 | 44.90 | 35.61 | $63.68_{(+9.14)}$ |
| Segmenter(Strudel et al., 2021) | T | 89.40 | 90.58 | 52.73 | 43.88 | 42.33 | $63.78_{(+9.04)}$ |
| Seq(Zhang et al., 2022) | T | 81.77 | 78.82 | 46.93 | 26.40 | 42.38 | $55.26_{(+17.56)}$ |
| ANNEXE (Su et al., 2025b) | L | 92.45 | 93.18 | 54.39 | 46.60 | 40.71 | $65.36_{(+7.46)}$ |
| CaRe-Ego (Su et al., 2025a) | T | **94.47** | 94.41 | 51.56 | 36.80 | 41.84 | $63.82_{(+9.00)}$ |
| **InerFormer (Ours)** | T | 94.38 | **94.87** | **66.79** | 55.79 | 52.25 | **72.82** |

we incorporate standard task-specific losses: the classification loss $\mathcal{L}_{cls}$, the dice loss $\mathcal{L}_{dic}$, and the mask cross-entropy loss $\mathcal{L}_{ce}$, to optimize class prediction and mask quality. The overall training objective is formulated as a weighted combination of these components:

$$\mathcal{L}_{co} = \mathcal{L}_{co}^{left} + \mathcal{L}_{co}^{right} + \mathcal{L}_{co}^{two}, \mathcal{L} = \lambda_b \cdot \mathcal{L}_b + \lambda_{co} \cdot \mathcal{L}_{co} + \lambda_{cls} \cdot \mathcal{L}_{cls} + \lambda_{dic} \cdot \mathcal{L}_{dic} + \lambda_{ce} \cdot \mathcal{L}_{ce}, \quad (10)$$

where the $\lambda_b, \lambda_{co}, \lambda_{cls}, \lambda_{dic}$, and $\lambda_{ce}$ are non-negative hyperparameters that balance the contributions of each loss term. These weights are kept fixed throughout training in our experiments.

## 4 EXPERIMENTS

### 4.1 DATASETS AND METRICS

To evaluate the effectiveness and generalization capability of our InterFormer, we conduct comprehensive comparisons with various state-of-the-art (SOTA) methods on the EgoHOS (Zhang et al., 2022) and mini-HOI4D (Su et al., 2025a) datasets.

**EgoHOS.** This dataset is an egocentric dataset with pixel-level annotations for hands and interacting objects, containing 8,993 *training*, 1,124 *validation*, 1,126 *in-domain test*, and 500 *out-of-domain test* image-mask pairs.

**mini-HOI4D.** This dataset is derived from the HOI4D dataset (Liu et al., 2022a), which consists of 1,095 egocentric images with corresponding mask annotations for hands and active objects. We use this dataset to evaluate the generalization ability of the proposed method under OOD conditions.

**Evaluation Metrics and Implementation Details.** We assess segmentation performance using standard metrics: (mean) Intersection-over-Union (IoU) and pixel accuracy (Acc). Due to space limitations, we present the IoU and mIoU results in this paper. The Acc results are provided in Appendix A.1. The implementation details are described in Sec. 8.

Table 3: Comparison with SOTA methods on the mini-HOI4D dataset measured by IoU ↑.

| Method | Type | Left Hand | Right Hand | Right-hand Object | Two-hand Object | Overall Object |
|---|---|---|---|---|---|---|
| Segformer(Xie et al., 2021) | T | 30.16 | 56.44 | 5.17 | 12.02 | $25.95_{(+40.12)}$ |
| SCTNet(Xu et al., 2024b) | T | 35.83 | 66.29 | 17.72 | 20.98 | $35.21_{(+30.86)}$ |
| Multi-UNet (Zhao et al., 2025) | C | 52.15 | 83.64 | 25.70 | 41.60 | $42.36_{(+23.71)}$ |
| UperNet(Xiao et al., 2018) | C | 54.82 | 84.43 | 20.34 | 29.34 | $47.23_{(+18.84)}$ |
| MaskFormer(Cheng et al., 2022a) | T | 58.50 | 83.66 | 35.28 | 56.91 | $58.59_{(+7.48)}$ |
| Segmenter(Strudel et al., 2021) | T | **74.70** | 85.58 | 22.38 | 58.67 | $60.33_{(+5.74)}$ |
| Seq(Zhang et al., 2022) | T | 8.74 | 34.60 | 23.88 | 53.96 | $30.30_{(+35.77)}$ |
| Mask2Former(Cheng et al., 2022a) | T | 70.13 | 88.57 | 32.37 | 55.72 | $61.70_{(+4.37)}$ |
| ANNEXE (Su et al., 2025b) | L | 68.06 | 85.13 | 40.93 | 57.36 | $62.87_{(+3.20)}$ |
| CaRe-Ego (Su et al., 2025a) | T | 70.39 | **89.76** | 27.56 | 60.08 | $61.95_{(+4.12)}$ |
| **InterFormer (Ours)** | T | 66.44 | 87.07 | **46.30** | **64.48** | **66.07** |

Figure 5: Visualization results on (a) EgoHOS in-domain test set, (b) EgoHOS out-of-domain test set, and (c) out-of-distribution mini-HOI4D dataset.

## 4.2 COMPARISONS WITH STATE-OF-THE-ART METHODS

We conduct a comprehensive in-domain and OOD comparison of the InerFormer with SOTA Ego-HOS models in Tables 1-3, including convolution-based (C), transformer-based (T), and large language model-based (L). The best results are in **bold**, and the second best is underlined.

### 4.2.1 IN-DOMAIN COMPARISON RESULTS

Table 1 presents a comparative analysis of InterFormer against methods on the EgoHOS in-domain benchmark. InterFormer achieves superior performance across all categories, attaining an impressive mIoU of 73.22%. The most pronounced advantage is observed in object segmentation, particularly for two-handed objects, where it achieves an outstanding IoU of 64.17%, representing a substantial improvement of 7.76% in IoU over the second-best method. This significant progress can be attributed to our interaction-centric design, which leverages DQG to adapt queries to diverse interacting objects, employs DFS to enhance the feature representation of interactions, and utilizes the CoCo loss to enforce robust hand-object correlations.

### 4.2.2 OUT-OF-DISTRIBUTION COMPARISON RESULTS.

**Evaluation on the EgoHOS out-of-domain test set.** To evaluate the generalization ability of our model, we assessed InerFormer on the out-of-domain EgoHOS test set using the best saved checkpoint, as shown in Table 2. Iner-Former achieved the highest overall mIoU of 72.82%, outperforming the second-best method by 7.46%. Notably, our approach also attained the highest IoU scores for all three categories of interacting objects.

**Evaluation on the mini-HOI4D dataset.** We also conducted OOD testing on the challenging

Table 4: Ablation study results on the EgoHOS in-domain test set.

| # | IPP | DQG | DFS | CoCo | Performance (%) mIoU ↑ | mAcc ↑ |
|---|---|---|---|---|---|---|
| 1 | – | – | – | – | 70.72 | 77.48 |
| 2 | – | – | – | ✓ | 70.95 | 79.02 |
| 3 | ✓ | – | – | – | 71.23 | 79.97 |
| 4 | ✓ | ✓ | – | – | 71.50 | 79.68 |
| 5 | ✓ | – | ✓ | – | 71.26 | 79.11 |
| 6 | ✓ | ✓ | ✓ | – | 72.35 | 80.13 |
| **Ours** | ✓ | ✓ | ✓ | ✓ | **73.22** | **80.68** |

mini-HOI4D dataset in Table 3. Our proposed InerFormer method achieved the highest mIoU of 66.07%, surpassing the second-best method by 3.20%. In conclusion, these results underscore the generalization ability of InerFormer in challenging out-of-domain settings, confirming that the proposed module can dynamically understand the interactive relationships between hands and objects.

## 4.3 Ablation Study

**Efficacy of InerFormer**. We present an ablation study to assess the contributions of InerFormer and its core components. For fair comparison, all experiments used identical model configurations. Since DQG and DFS are built upon the IPP Branch, we also ablate this branch. Table 4 summarizes the results, leading to three key observations: 1) The IPP branch significantly improves performance, as it can localize hand-object interaction regions. 2) Each additional component brings further incremental improvements. 3) Integrating all components, the complete InerFormer framework achieves the best performance, proving the effectiveness of the framework and its individual components.

**Hyperparameter study**. In CoCo loss, the threshold $\tau$ determines that a class is considered present only if the predicted pixel number exceeds $\tau$. To assess its impact, we varied $\tau$ from 50 to 300 in steps of 50 in Table 5. Performance peaks at $\tau = 100$, consistent with theoretical expectations: when $\tau$ is too small, the model becomes overly sensitive, yielding spurious hand detections (false positives). Conversely, when $\tau$ is too large, the model may miss partially visible hands (false negatives). Thus, this trade-off explains the observed optimum at $\tau = 100$. Due to page limits, analysis of $\lambda_b, \lambda_{co}, \lambda_{cls}, \lambda_{dice}$, and $\lambda_{ce}$ is presented in Appendix A.2.

Table 5: Hyperparameter experiments of $\tau$ on the EgoHOS in-domain test set.

| # | Hyper Parameter $\tau$ | Performance mIoU | mAcc |
|---|---|---|---|
| 1 | 50 | 71.62 | 80.62 |
| 2 | 100 | **73.22** | **80.68** |
| 3 | 150 | 71.97 | 80.51 |
| 4 | 200 | 71.62 | 80.94 |
| 5 | 250 | 71.10 | 78.78 |
| 6 | 300 | 72.38 | 80.12 |

## 4.4 Visualization Results.

For visual comparison, we show results against the CaRe-Ego baseline (Su et al., 2025a). Fig. 5(a)-(c) demonstrates that our method achieves superior segmentation of interacting objects by explicitly modeling interaction-aware representations. More visualization results can be found in Appendix.

## 5 Conclusion

We propose a novel InerFormer approach for the EgoHOS task. Specifically, we introduce the DQG module to create robust queries that can adapt to various interacting objects in different images. We further design the DFS module to encourage the network to explicitly perceive interaction-aware features. Additionally, our CoCo loss guides the network to learn interaction relationships that are consistent with real-world logic. Experimental results on three public test sets demonstrate the remarkable effectiveness and generalization of InerFormer.

## 6 Acknowledgements

The research work described in this paper was conducted in the JC STEM Lab of Machine Learning and Computer Vision funded by The Hong Kong Jockey Club Charities Trust. This research received partially support from the Global STEM Professorship Scheme from the Hong Kong Special Administrative Region.

## 7 Ethics statement

Our research work fully adheres to the ICLR Code of Ethics and embodies its core principles through the following commitments: 1) We maintain the highest standards of scientific rigor by providing comprehensive experimental results, detailed methodology descriptions, and open-source code implementation to ensure full reproducibility and verification of our findings. 2) As fundamental algorithmic research, our work contributes to the advancement of egocentric vision without targeting any specific application domain that might raise ethical concerns. The proposed techniques are designed to be generally applicable across diverse contexts. 3) Our research utilizes only publicly available benchmark datasets with proper licensing, ensuring no privacy violations or discriminatory impacts.

The algorithmic improvements focus on technical efficiency without embedding biases against any demographic groups. 4) We faithfully acknowledge all referenced works and properly cite prior research contributions. Our comparisons with existing methods are conducted fairly under standardized evaluation protocols. 5) By open-sourcing our code and models, we promote equitable access to our research outcomes and encourage broader community participation in further development.

## 8 REPRODUCIBILITY STATEMENT

All experiments were carried out on four NVIDIA RTX 4090 GPUs, using a total batch size of 8. During data preparation, images were cropped to 448×448 pixels and normalized using a mean of [106.011, 95.400, 87.429] and a standard deviation of [64.357, 60.889, 61.419]. The maximum number of training iterations was set to 180k. Following previous methods (Cheng et al., 2022a), The number of queries is set to the number of target categories, i.e., 5. The values of $\lambda_b$, $\lambda_{co}$, and $\lambda_{cls}$ were set to 1, while $\lambda_{ce}$ and $\lambda_{dic}$ were set to 5. The hyperparameter $\tau$ in CoCo loss is set to 100. The model was trained end-to-end using the AdamW optimizer with an initial weight decay of 0.01. More implementation details can be found in the Appendix. B. The learning rate followed a two-phase schedule: it was linearly warmed up from 0 to 1e-4 over the first 10,000 iterations, and then decayed polynomially to zero by the end of training.

To ensure the reproducibility of our results and promote further research in the community, we will release all code, models, and implementation details upon paper acceptance. The repository will include comprehensive documentation, training scripts, and inference instructions to facilitate easy adoption and validation of our method.

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

# A APPENDIX

In this supplementary material, we first introduce the comparison experimental results of our Inter-Former against other SOTA methods measured by Acc in Sec. A.1. The implementation details are introduced in Sec. 8. Finally, the hyperparameter experiments of loss weights are described in Sec. A.2.

## A.1 COMPARISONS WITH STATE-OF-THE-ART METHODS

We conduct a comprehensive comparison of the proposed InterFormer with SOTA egocentric hand-object segmentation models, including convolution-based (C), transformer-based (T), and large language model-based (L) methods. This evaluation is performed on the EgoHOS in-domain test set. Furthermore, to assess the generalization capability of our approach, we evaluate all methods using their best saved training checkpoints on the EgoHOS out-of-domain test set and the mini-HOI4D dataset. We exhibit the comparison results on three test sets using IoU in the main paper, so we add the comparison results on three test sets using Acc in this supplementary material.

### A.1.1 IN-DOMAIN COMPARISON RESULTS

Table 6: Comparison with SOTA methods on the EgoHOS in-domain test set measured by Acc ↑ (%) mAcc ↑ (%). The best results are in **bold** and the second best is underlined. T: transformer-based methods, C: convolution-based methods, L: MLLM-based methods.

| Method | Type | Left Hand ↑ | Right Hand ↑ | Left-hand Objects ↑ | Right-hand Objects ↑ | Two-hand Objects ↑ | Overall mAcc ↑ |
|---|---|---|---|---|---|---|---|
| Segformer (Xie et al., 2021) | T | 75.47 | 78.13 | 4.57 | 3.17 | 5.57 | 33.38 |
| SCTNet (Xu et al., 2024b) | T | 90.25 | 89.92 | 24.49 | 20.79 | 29.08 | 50.91 |
| Para (Zhang et al., 2022) | T | 75.57 | 75.93 | 39.50 | 39.33 | 42.58 | 54.58 |
| Segmenter (Strudel et al., 2021) | T | 89.87 | 91.92 | 62.69 | 45.59 | 62.78 | 70.57 |
| UperNet(Xiao et al., 2018) | T | 89.86 | 91.32 | 37.24 | 42.26 | 49.27 | 61.99 |
| Multi-UNet (Zhao et al., 2025) | C | 89.01 | 91.37 | 60.71 | 43.76 | 48.35 | 66.64 |
| MaskFormer (Cheng et al., 2022a) | T | 95.90 | 96.41 | 67.08 | 52.91 | 64.86 | 75.43 |
| OneFormer(Zhang et al., 2022) | T | 96.21 | 96.33 | 64.19 | 53.06 | 63.75 | 74.71 |
| Mask2Former (Cheng et al., 2022a) | T | 96.01 | 96.20 | 53.97 | 58.10 | 60.48 | 72.95 |
| ANNEXE (Su et al., 2025b) | L | 95.87 | 94.81 | 73.28 | 66.54 | 68.50 | 79.80 |
| Seq (Zhang et al., 2022) | T | 95.77 | 91.29 | 66.67 | 59.85 | 62.21 | 75.16 |
| Care-Ego (Su et al., 2025a) | T | **96.64** | **96.81** | 71.79 | **68.71** | 65.85 | 79.96 |
| **InterFormer (Ours)** | T | 96.58 | 96.55 | **74.06** | 65.93 | **70.26** | **80.68** |

Table 7: Comparison with SOTA methods on the EgoHOS out-of-domain test set using Acc ↑ (%) mAcc ↑ (%). The best results are in **bold**, and the second best is underlined. T: transformer-based methods, C: convolution-based methods, L: MLLM-based methods.

| Method | Type | Left Hand | Right Hand | Left-hand Object | Right-hand Object | Two-hand Object | Overall |
|---|---|---|---|---|---|---|---|
| Segformer(Xie et al., 2021) | T | 86.80 | 79.44 | 21.77 | 9.56 | 11.67 | 41.85 |
| SCTNet(Xu et al., 2024b) | T | 94.62 | 90.92 | 49.14 | 27.47 | 17.12 | 55.85 |
| UperNet(Xiao et al., 2018) | C | 96.89 | 96.00 | 64.83 | 54.59 | 27.83 | 68.03 |
| Multi-UNet (Zhao et al., 2025) | C | 94.75 | 95.80 | 45.16 | 47.89 | 16.75 | 60.07 |
| Maskformer(Cheng et al., 2022a) | T | 95.58 | 96.10 | 70.53 | 60.49 | 46.52 | 73.84 |
| Mask2former(Cheng et al., 2022a) | T | 97.05 | 96.38 | 64.39 | 64.18 | 39.78 | 72.36 |
| Seq(Zhang et al., 2022) | T | 87.83 | 85.98 | 57.17 | 43.85 | 54.76 | 65.92 |
| ANNEXE (Su et al., 2025b) | L | 97.03 | 96.80 | 76.57 | 60.35 | 52.35 | 76.62 |
| CaRe-Ego (Su et al., 2025a) | T | 97.09 | 96.69 | 72.30 | 60.90 | 46.28 | 74.65 |
| **InterFormer (Ours)** | T | **97.21** | **97.10** | **79.22** | **68.19** | **58.74** | **80.09** |

Table 6 presents a comparative analysis of InterFormer against leading convolution-based, transformer-based, and large language model-based methods on the EgoHOS in-domain benchmark. As shown, InterFormer achieves superior performance across all categories, attaining an impressive

mAcc of 80.68%. While InterFormer excels in hand segmentation, its most pronounced advantage is observed in object segmentation, particularly for left-hand objects and two-handed objects, where it achieves an outstanding Acc of 74.06% and 70.26%, respectively. This significant progress can be attributed to our interaction-centric design, which leverages DQG to adapt queries to diverse interacting objects, employs DFS to enhance the feature representation of interactions, and utilizes the CoCo loss to enforce robust hand-object correlations.

### A.1.2 OUT-OF-DISTRIBUTION COMPARISON RESULTS

**Comparison on EgoHOS out-of-domain test set measured by Acc.** To thoroughly evaluate the generalization capability of our model, we conducted an assessment of InterFormer on the out-of-domain EgoHOS test set, utilizing the best saved checkpoint for this purpose. The results of this evaluation are presented in Table 7. Notably, InterFormer achieved an impressive overall accuracy score of 80.09%, which enables it to outperform the second-best method by a margin of 3.47%. This achievement is particularly significant, as our approach not only excelled overall but also secured the highest Acc scores across all categories of hands and interacting objects. These findings strongly indicate the robust generalization ability of our InterFormer method across diverse scenarios.

Table 8: Comparison with SOTA methods on the mini-HOI4D dataset measured by Acc ↑ (%) mAcc ↑ (%). The best results are in **bold**, and the second best is underlined. T: transformer-based methods, C: convolution-based methods, L: MLLM-based methods.

| Method | Type | Left Hand | Right Hand | Right-hand Object | Two-hand Object | Overall Object |
|---|---|---|---|---|---|---|
| Segformer(Xie et al., 2021) | T | 92.13 | 72.42 | 5.52 | 13.41 | 45.87 |
| SCTNet(Xu et al., 2024b) | T | 95.25 | 71.27 | 22.68 | 29.98 | 54.90 |
| Multi-UNet (Zhao et al., 2025) | C | 94.37 | 70.25 | 30.41 | 47.65 | 60.67 |
| UperNet(Xiao et al., 2018) | C | 97.71 | 86.04 | 25.77 | 36.58 | 61.53 |
| MaskFormer(Cheng et al., 2022a) | T | 96.63 | 87.83 | 44.81 | 73.47 | 75.69 |
| Seq(Zhang et al., 2022) | T | 40.90 | 38.05 | 28.99 | 61.67 | 42.40 |
| Mask2Former(Cheng et al., 2022a) | T | 97.48 | 89.38 | 45.22 | 74.17 | 76.56 |
| ANNEXE (Su et al., 2025b) | L | 96.54 | **93.18** | 48.77 | **75.19** | 78.42 |
| CaRe-Ego (Su et al., 2025a) | T | **97.79** | 93.09 | 49.35 | 68.01 | 77.06 |
| **InterFormer (Ours)** | T | 96.55 | 91.71 | **59.75** | 74.70 | **80.68** |

**Comparison on mini-HOI4D dataset measured by Acc.** In addition to the EgoHOS test set, we also performed out-of-distribution testing on the challenging mini-HOI4D dataset, the results of which are summarized in Table 8. Our proposed InterFormer method demonstrated its efficacy by attaining the highest mean Accuracy (mAcc) of 80.68%, thereby surpassing the second-best method by 2.26%. These results further prove the generalization ability of InterFormer in demanding out-of-domain settings. They confirm that the proposed module is adept at dynamically understanding and interpreting the interactive relationships between hands and objects, showcasing its potential for real-world applications in egocentric scenarios.

Table 9: Comparison of computational complexity and performance on EgoHOS test set

| Method | Type | FLOPs | mIoU (%) |
|---|---|---|---|
| SegFormer | Transformer-based | 71.961G | 27.89 |
| Segmenter | Transformer-based | 70.074G | 59.52 |
| Mask2Former | Transformer-based | 96.093G | 64.88 |
| Seq | Transformer-based | 392.483G | 67.17 |
| ANNEXE | MLLM-based | 610.500G | 71.38 |
| **InterFormer (Ours)** | Transformer-based | **122.996G** | **73.22** |

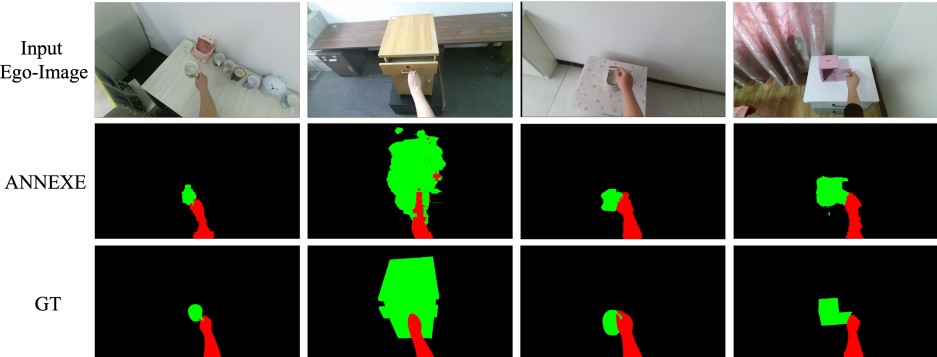

Figure 6: The MLLM-based methods (ANNEXE) show limited spatial precision in mask generation for parsing hands and interacting objects.

### A.1.3 COMPLEXITY OF INTERFORMER

Figure 1 compares the model sizes of different methods. It shows that our InterFormer achieves state-of-the-art performance with a manageable increase in parameters, effectively trading a compact model structure for superior accuracy.

To evaluate computational complexity, we present a comparative analysis of our InterFormer against other methods in terms of FLOPs. The results (Tab. 9) are based on testing conducted on the Ego-HOS in-domain test set. These results demonstrate that our method achieves a favorable balance between FLOPs and mean Intersection over Union (mIoU).

### A.1.4 ERROR ANALYSIS: INTERFORMER VS. MLLMS.

To better understand the comparative strengths and limitations of our InterFormer versus Multimodal Large Language Models (MLLMs), such as ANNEXE, we conducted a detailed failure mode analysis.

**Spatial Precision in Mask Generation.** Our analysis confirms that MLLMs consistently produce masks with coarse and inaccurate boundaries. As shown in Figure 6, MLLM-generated masks (ANNEXE row) can classify the categories of each predicted entity. However, the predicted masks for hands and objects often exhibit poor alignment with true edges, which is caused by the lack of interaction-centric representation learning. In contrast, our proposed InterFormer is specifically designed for egocentric hands and active object parsing, enabling the generation of more precise masks with clear boundaries.

**Prompt Sensitivity.** All evaluated MLLMs require text prompts and show high sensitivity to prompt design. In our experiments, we used a detailed prompt specifying five distinct mask types (left/right hands, corresponding interacting objects, and two-hand objects). However, ANNEXE struggled to understand and generate the required masks under this instruction, revealing limitations in following detailed segmentation tasks.

### A.2 HYPERPARAMETER STUDY

The InterFormer framework is supervised by the interaction boundary loss $\mathcal{L}_b$, CoCo loss $\mathcal{L}_{co}$, the classification loss $\mathcal{L}_{cls}$, dice loss $\mathcal{L}_{dic}$, and mask loss $\mathcal{L}_{ce}$ for model training. The overall loss function is defined as:

$$\mathcal{L}_{co} = \mathcal{L}_{co}^{left} + \mathcal{L}_{co}^{right} + \mathcal{L}_{co}^{two}, \tag{11}$$

$$\mathcal{L} = \lambda_b \cdot \mathcal{L}_b + \lambda_{co} \cdot \mathcal{L}_{co} +$$
$$\lambda_{cls} \cdot \mathcal{L}_{cls} + \lambda_{dic} \cdot \mathcal{L}_{dic} + \lambda_{ce} \cdot \mathcal{L}_{ce}, \tag{12}$$

Table 10: Hyperparameter study of different loss weights on the EgoHOS in-domain test set. The best results are shown in **bold**.

| # | $\lambda_b$ | $\lambda_{co}$ | $\lambda_{cls}$ | $\lambda_{dic}$ | $\lambda_{ce}$ | Performance (%) | |
|---|---|---|---|---|---|---|---|
| | | | | | | mIoU ↑ | mAcc ↑ |
| 1 | 1 | 1 | 1 | 1 | 1 | 71.74 | 78.40 |
| 2 | 5 | 1 | 1 | 1 | 1 | 71.79 | 78.21 |
| 3 | 1 | 5 | 1 | 1 | 1 | 71.88 | 78.59 |
| 4 | 5 | 5 | 1 | 1 | 1 | 71.08 | 79.71 |
| 5 | 1 | 1 | 5 | 1 | 1 | 71.30 | 78.79 |
| 6 | 5 | 5 | 1 | 5 | 5 | 72.08 | 79.71 |
| **Ours** | 1 | 1 | 1 | 5 | 5 | **73.22** | **80.68** |

where the $\lambda_b, \lambda_{co}, \lambda_{cls}, \lambda_{dic}$, and $\lambda_{ce}$ are hyperparameters used to balance the contributions of each loss item. In this section, we conducted hyperparameter experiments to verify the impact of different loss weights on the EgoHOS in-domain test set. To ensure experimental fairness, all other parameters except the loss weight were the same as those described in Section 8. The experimental results are shown in Table 10. Our model achieved the highest mean intersection over union (mIoU) of 73.22% and mean average accuracy (mAcc) of 80.68% with the hyperparameter configurations of $\lambda_b = 1, \lambda_{co} = 1, \lambda_{cls} = 1, \lambda_{dic} = 5$, and $\lambda_{ce} = 5$. This result demonstrates that appropriate loss weights can significantly improve model performance. Experimental results highlight the importance of balancing the weights of different loss functions in our model.

## A.3 EFFECTIVENESS OF COCO LOSS

**Qualitative results.** To specifically validate the contribution of the CoCo loss, Figure 7 demonstrates the qualitative improvements achieved by incorporating this component. The visualization contrasts model predictions without ("w/o coco") and with ("w/ coco") the CoCo loss across two datasets. Without the CoCo loss, the model erroneously predicts objects as interacting with the left hand even when no left hand is present. In contrast, incorporating the CoCo loss substantially enhances the physical consistency of hand-object interactions in the predictions.

Table 11: Quantitive results of CoCo loss.

| # | | Method | Rate ↓ |
|---|---|---|---|
| 1 | | Seq Zhang et al. (2022) | 9.80% |
| 2 | | Care-Ego Su et al. (2025a) | 5.45% |
| 3 | | Ours w/o CoCo | 2.19% |
| 4 | | Ours w/ CoCo | 1.55% |

**Quantitative results.** We conducted a systematic evaluation to quantify the effectiveness of the CoCo loss in mitigating the interaction illusion problem. As summarized in Table 11, we measured the frequency of this phenomenon by calculating the percentage of predictions containing interaction illusions across all outputs generated by our model. The results demonstrate that our model trained without the CoCo loss produces such artifacts in 2.19% of its predictions, whereas incorporating the CoCo loss ("w/ CoCo") reduces this rate to 1.55%, which represents a significant reduction of 0.64%. This quantitative evidence strongly confirms the critical role of the CoCo loss in suppressing spurious non-interactive object segments.

## A.4 VISUALIZATION RESULTS

This section presents comparative visualizations between our method and baseline approaches in Figure 8. Additionally, Figure 9 showcases representative failure cases that reveal the current limitations of our InterFormer framework. Based on the presented failure cases, we observe that our method tends to miss small objects or overlook parts of objects with significant appearance variations during parsing.

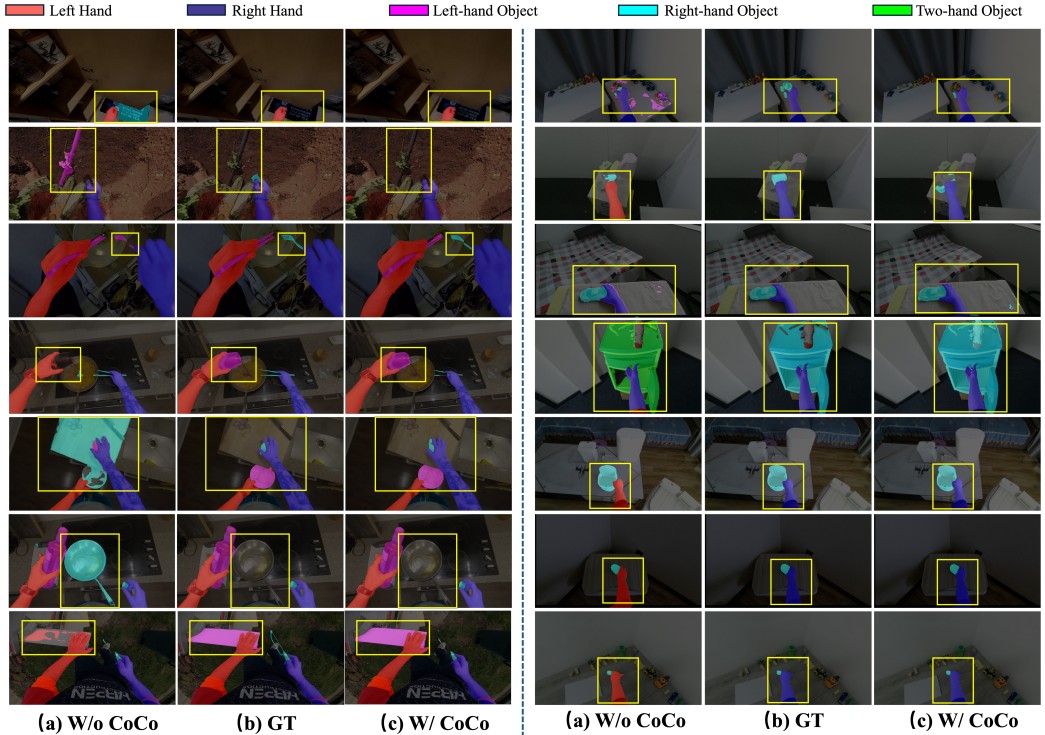

Figure 7: Qualitative comparison with/without employing CoCo Loss on EgoHOS (left) and mini-HOI4D (right) datasets.

## A.5 LIMITATIONS AND FUTURE WORK

### A.5.1 LIMITATIONS.

**Limitations of hand-object interaction understanding in egocentric images.** One significant limitation is that in more realistic and complex scenarios, a hand may not be visible in a specific frame, yet it can still actively engage in interactions between objects. This situation underscores the challenge of relying solely on static images for understanding dynamic interactions.

**Limitations about pixel count in CoCo loss.** The proposed CoCo loss relies on an absolute pixel threshold $\tau$ to determine the presence of hands. Although simple and efficient, it introduces sensitivity to imaging conditions such as hand-camera distance and image resolution.

**Occulion.** While the proposed method demonstrates strong performance, it shares a common limitation with existing approaches in handling hands and objects that are heavily occluded. Our Inter-Former does not incorporate explicit mechanisms for occlusion reasoning or recovery. In scenarios where hand-object interactions are significantly obscured (e.g., by other objects or self-occlusion), the model may struggle to accurately segment boundaries or infer interaction contexts.

### A.5.2 FUTURE WORK.

**Hand-object interaction understanding in multi-view and/or video**. Our future work will focus on extending egocentric hand-object segmentation to encompass both video and multi-view scenarios. By integrating more information, we aim to develop a more robust framework that can accurately capture interactions, even when the hands are occluded or not visible in specific frames. This advancement will not only enhance the reliability of hand-object interaction detection but also expand the applicability of our research to fields such as robotics, augmented reality, and human-computer interaction.

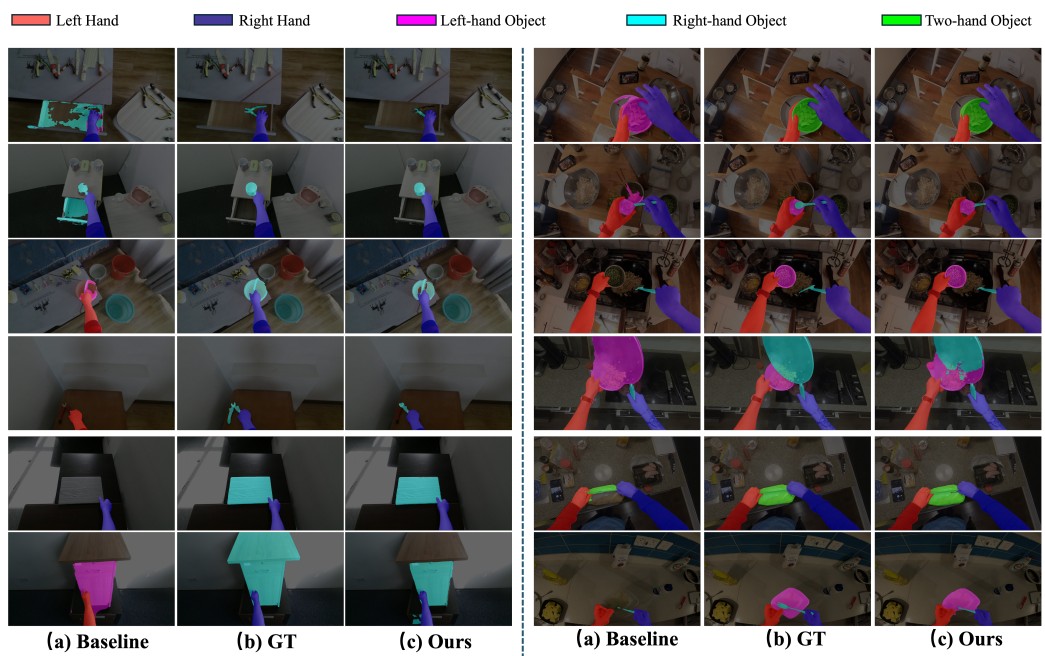

Figure 8: Qualitative comparison of our InterFormer against baseline on OOD mini-HOI4D (left) and EgoHOS out-of-domain (right) test sets.

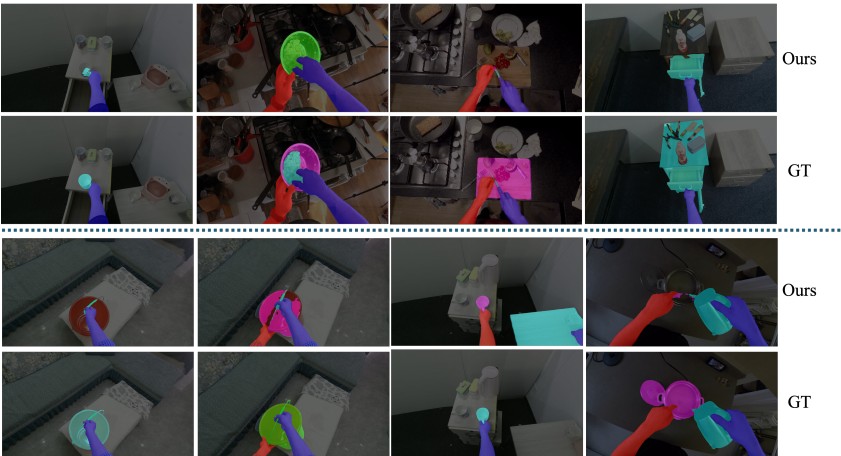

Figure 9: Failure cases of our method.

**Replace the pixel count in CoCo loss.** In future work, we plan to explore adaptive thresholding mechanisms to replace the pixel count in CoCo loss to improve generalization. A direction is to normalize the pixel count of hands, making the threshold condition relative to the visual scene. Additionally, we will investigate learning-based approaches where the presence threshold is dynamically determined by the network rather than pre-defined.

**Deploying and evaluating on AR/MR devices.** Evaluating the model's performance in dynamic real-world settings will provide valuable insights into its robustness and adaptability, further informing improvements. We anticipate that this research could significantly advance the state of the art in interactive applications, paving the way for more intuitive and engaging user experiences in AR and MR technologies.

**Addressing the occlusion problem.** Future work will explore dedicated strategies to address this challenge, such as introducing occlusion-aware attention mechanisms, leveraging temporal consistency in video sequences, or integrating generative models to reconstruct plausible structures in occluded regions. We believe these directions will further enhance the robustness of egocentric hand-object interaction analysis in real-world settings.

## A.6 THE USE OF LARGE LANGUAGE MODELS (LLMS)

Large language models (LLMs) were used exclusively for language refinement and proofreading during the preparation of this manuscript. The model assisted in improving grammar, clarity, and overall readability of the text without altering the original technical content, data, or scientific conclusions. All research ideas, analysis, and writing were conducted by the authors; the model was not involved in any aspect of content generation or decision-making.

