# OpenReview forum: "Interaction-aware Representation Modeling With Co-Occurrence Consistency for Egocentric Hand-Object Parsing"
_ICLR.cc/2026/Conference — ICLR 2026 Poster_

### Official Review · Reviewer_NB12 · 2025-10-26

**Soundness:** 2
**Presentation:** 2
**Contribution:** 2
**Rating:** 2
**Confidence:** 3

**Summary:**

This paper presents InterFormer, a transformer-based model for egocentric hand–object segmentation. It introduces three key components, Prototypical Query Generator (PQG), Dual-context Feature Selector (DFS), and Conditional Co-occurrence (CoCo) loss to improve interaction-aware representation and reduce physically inconsistent predictions. The model achieves state-of-the-art results on EgoHOS and mini-HOI4D datasets.

**Strengths:**

1. This paper provides clear figures to clarify the motivation and methodology.
2. The method achieves competitive performance on the reported benchmarks.

**Weaknesses:**

1. Limited novelty and over-engineered design. The paper mainly achieves performance gains through stacking multiple existing components, relying on a cascaded U-Net-style decoder to generate priors and a sequence of post-processing steps. The overall method appears as an incremental engineering integration rather than a conceptually novel contribution.
2. Inconsistent motivation and contribution. The authors claim that existing query initialization methods “lack adaptability to diverse categories of contacting objects,” and thus propose the Prototypical Query Generator (PQG). However, PQG only guides the network toward hand–object contact regions via similarity-based subregion selection and has little to do with modeling category diversity.
3. Misleading use of the term “Prototypical”. The proposed Prototypical Query Generator (PQG) constructs queries by combining interaction priors, subregion Top-N selection, and a learnable bias，yet the connection to 'prototypical' concepts appears tenuous.
4. Lack of model efficiency analysis. The method adds multiple stages and operations (e.g., Top-K feature selection), yet provides no analysis or comparison of model efficiency in terms of computational cost, or inference speed.
5. Insufficient diversity of datasets. The experiments are mainly conducted on mini-HOI4D and EgoHOS, which are highly similar in structure and data distribution. Incorporating additional datasets such as Ego-IRGBench[1] and VISOR-NVOS[2] is recommended.
[1] ANNEXE: Unified Analyzing, Answering, and Pixel Grounding for Egocentric Interaction. CVPR, 2025.
[2] Learning to segment referred objects from narrated egocentric videos. CVPR, 2024.
6. Poor writing quality. For instance, the notation system is inconsistent (L362), the references to Leonardi et al. (2024) and EgoHOS (L44) are unrelated, the citation for mini-HOI4D (L383) appears incorrect, and the claim "most existing methods" (L206) lacks supporting citations.
7. Limited practical applicability and broader impact. While the study is technically well-executed, its potential influence on future research directions appears modest.

**Questions:**

See weakness.

---

> ### Author Response · Authors · 2025-11-29
> **Author response to Reviewer NB12 (Weakness 1-2)**
>
> **Q1**: Limited novelty and over-engineered design. The paper mainly achieves performance gains through stacking multiple existing components, relying on a cascaded U-Net-style decoder to generate priors and a sequence of post-processing steps. The overall method appears as an incremental engineering integration rather than a conceptually novel contribution.
>
> **A1**:
>
> We thank the reviewer for their valuable feedback on the novelty and contribution of our work.
>
> The proposed InterFormer is not a simple composition of existing components. We design three novel modules, DQG, DFS, and CoCo loss, to address specific challenges in hand-object interaction parsing. Specifically, based on the coarse boundary-guided features from IPP, the DQG generates interaction-aware queries by fusing boundary-aligned semantic embeddings with learnable parameters. The DFS introduces an interaction-centric refinement mechanism that purifies semantic embeddings through boundary-guided feature fusion. The CoCo loss incorporates hand-object physical constraints to guide the model in perceiving interaction relationships.
>
> These modules are efficient yet effective, with each targeting a clear limitation in existing methods:
>
> * The DQG enables robust query adaptation to varying active objects across different images.
>
> * The DFS suppresses interaction-irrelevant noise and constructs discriminative interaction embeddings.
>
> * The CoCo loss specifically mitigates interaction illusion, ensuring physically consistent and accurate segmentation.
>
> The performance improvements observed across multiple benchmarks are the direct outcome of this principled design, rather than merely stacking components.
>
> By integrating all these modules, the proposed InterFormer demonstrates the effectiveness in the complex task of hand-object interaction parsing, representing a meaningful advancement. As highlighted by another reviewer, "*The idea to explicitly model interaction for extract hand-object segmentation masks under interaction is highly intuitive*." This underscores the fundamental novelty of our paradigm, i.e., shifting from traditional semantic-driven segmentation to interaction-aware parsing.
>
> We have revised the manuscript to better clarify the conceptual motivation and the synergistic role of each component within the overall framework.
>
>
> **Q2**: Inconsistent motivation and contribution. The authors claim that existing query initialization methods “lack adaptability to diverse categories of contacting objects,” and thus propose the Prototypical Query Generator (PQG). However, PQG only guides the network toward hand–object contact regions via similarity-based subregion selection and has little to do with modeling category diversity.
>
> **A2**:
>
> We sincerely thank the reviewer for this insightful comment, which helps us clarify a key point of confusion in our original manuscript. We fully understand the concern and apologize for the lack of precision in our initial motivation statement.
>
> Our original phrasing, "lack adaptability to diverse categories of contacting objects", was not intended to claim that PQG recognizes or distinguishes object categories. Instead, we aimed to emphasize that PQG addresses a more fundamental issue: existing methods often rely on learnable parameters or generic feature sampling to initialize queries, which lack a mechanism to focus on interaction, i.e., **identifying which objects are actively involved in an interaction in a given scene, regardless of their category.**
>
> Based on the above problem, we observe that despite the wide variation in object categories (e.g., cups, drawers, books), **interacting objects share a common visual property: spatial contact with hands**. Instead of modeling object categories explicitly, we leverage this more universal and stable cue to design the PQG.
>
> The PQG uses this category-agnostic interaction cue (spatial contact with hands) via similarity-based subregion selection to guide query initialization toward regions of hand–object contact. This allows the model to dynamically adapt to different interactive scenarios across images, focusing on where interaction occurs rather than what the object categories are.
>
> We have thoroughly revised the motivation and contribution sections in the manuscript to articulate this logic more clearly and accurately.

---

> > ### Author Response · Authors · 2025-11-29
> > **Author response to Reviewer NB12 (Weakness 3-5)**
> >
> > **Q3**: Misleading use of the term “Prototypical”. The proposed Prototypical Query Generator (PQG) constructs queries by combining interaction priors, subregion Top-N selection, and a learnable bias，yet the connection to 'prototypical' concepts appears tenuous.
> >
> > **A3**:
> >
> > We thank the reviewer for raising this valid concern regarding the use of the term "Prototypical" in our original naming. We acknowledge this oversight and apologize for any conceptual ambiguity it may have caused.
> >
> > Our initial intention in naming it "Prototypical Query Generator" was to highlight that the module generates queries that serve as representative feature vectors for each target interaction category (e.g., left hand, right hand, interacting objects) within the given input image. In this context, each query was intended to function as a "prototype" for one of the predefined categories in the current scene.
> >
> > However, we fully agree with the reviewer that the term "prototypical" in computer vision is more conventionally tied to class-level representations learned across multiple samples, often through clustering or metric learning. Since our method does not involve cross-sample prototype learning and relies solely on intra-image feature selection, we recognize that the original naming was misleading.
> >
> > To more accurately reflect the module's function, i.e., dynamically generating queries based on interaction-guided feature selection, we have renamed it the Dynamic Query Generator (DQG) in the revised manuscript and removed all references to "prototypical." We believe this change aligns better with established terminology and more clearly conveys the module's purpose.
> >
> > Thank you again for this valuable feedback, which has helped enhance the clarity and precision of our work.
> >
> > **Q4**: Lack of model efficiency analysis. The method adds multiple stages and operations (e.g., Top-K feature selection), yet provides no analysis or comparison of model efficiency in terms of computational cost, or inference speed.
> >
> > **A4**:
> >
> > We thank the reviewer for raising this important point regarding model efficiency. We agree that an analysis of computational cost is crucial.
> >
> > In our revised manuscript, we have included a comprehensive comparison of FLOPs (Table 9 in Page 16) as a key metric for computational complexity. The results are also presented in Table below for convenience. These results demonstrate that our method achieves a superior balance between efficiency and performance: it attains the highest mIoU while maintaining a computational cost significantly lower than other high-performing methods like Seq and ANNEXE.
> >
> > |      Method      |       Type       |   FLOPs   |  mIoU  |
> > |:----------------:|:----------------:|:---------:|:------:|
> > |    SegFormer     | Transformer-based|  71.961G  | 27.89  |
> > |    Segmenter     | Transformer-based|  70.074G  | 59.52  |
> > |   Mask2Former    | Transformer-based|  96.093G  | 64.88  |
> > |       Seq        | Transformer-based|  392.483G | 67.17  |
> > |      ANNEXE      |    MLLM-based    |  610.500G | 71.38  |
> > | InterFormer (Ours) | Transformer-based| 122.996G | 73.22  |
> >
> >
> > **Q5**: Insufficient diversity of datasets. The experiments are mainly conducted on mini-HOI4D and EgoHOS, which are highly similar in structure and data distribution. Incorporating additional datasets such as Ego-IRGBench[1] and VISOR-NVOS[2] is recommended. [1] ANNEXE: Unified Analyzing, Answering, and Pixel Grounding for Egocentric Interaction. CVPR, 2025. [2] Learning to segment referred objects from narrated egocentric videos. CVPR, 2024.
> >
> > **A5**:
> >
> > We appreciate the reviewer’s suggestion regarding dataset diversity.
> >
> > However, we want to clarify that the datasets we used, mini-HOI4D and EgoHOS, are **in fact quite distinct in terms of data sources, scene settings, and task challenges**.
> >
> > * For the data source, the EgoHOS is constructed from multiple public datasets, including Ego4D, THU-READ, EPIC-KITCHENS, and additionally includes self-collected real-world egocentric videos. In contrast, mini-HOI4D is derived solely from the HOI4D dataset.
> >
> > * For scene and object diversity, the EgoHOS contains scenes from both indoor (e.g., kitchens) and outdoor environments, often with cluttered and dynamic backgrounds. On the other hand, mini-HOI4D is exclusively indoor and includes deliberately placed distractor objects that are visually similar to the active objects, making the task of identifying the true interactive object more challenging.
> >
> > While we acknowledge the value of Ego-IRGBench and VISOR-NVOS, these datasets are not directly applicable to our task:
> >
> > * The mask in Ego-IRGBench corresponds to the input text prompt, unlike our approach of directly segmenting all entities in interaction.
> >
> > * VISOR-NVOS is designed for video-level object segmentation and does not align with our image-based hand-object interaction segmentation setting.
> >
> > We hope these explanations solve your confusion.

---

> > > ### Author Response · Authors · 2025-11-29
> > > **Author response to Reviewer NB12 (Weakness 6,7)**
> > >
> > > **Q6**: Poor writing quality. For instance, the notation system is inconsistent (L362), the references to Leonardi et al. (2024) and EgoHOS (L44) are unrelated, the citation for mini-HOI4D (L383) appears incorrect, and the claim "most existing methods" (L206) lacks supporting citations.
> > >
> > > **A6**:
> > >
> > > We sincerely thank the reviewer for their valuable feedback on the writing quality of our paper. We apologize for the inconsistencies and inaccuracies in the original manuscript.
> > >
> > > In the revised version, we have thoroughly addressed all the issues raised:
> > >
> > > 1.	Notation inconsistency (e.g., L362): We have systematically reviewed and unified all mathematical notations throughout the paper.
> > >
> > > 2.	Inappropriate citations: The irrelevant reference to Leonardi et al. (2024) has been removed. The citation of EgoHOS (L44) has been corrected to ensure relevance. The reference to mini-HOI4D (L383) has been verified and updated.
> > >
> > > 3.	Unsupported claims: Statements such as "most existing methods" (L206) have been supported with appropriate references.
> > >
> > > Additionally, we have polished the language word-for-word and improved the overall logic and readability of the manuscript. We believe that the fully revised manuscript has significantly improved in writing quality. We thank the reviewers again for your rigorous and constructive review, which was crucial to enhancing the quality of our work.
> > >
> > > **Q7**:  Limited practical applicability and broader impact. While the study is technically well-executed, its potential influence on future research directions appears modest.
> > >
> > > **A7**:
> > >
> > > We thank the reviewer for the valuable feedback regarding the broader impact and practical applicability. We have carefully considered this point and have revised the manuscript to more clearly articulate the potential influence of our study, both methodologically and in terms of real-world applications.
> > >
> > > **1）Methodological Impact.** Our work introduces a novel interaction-aware representation learning paradigm to egocentric vision. The three core components we propose—PQG, DFS, and the CoCo loss—systematically address the existing problems. This establishes a clear technical framework for shifting from passively recognizing “what it is” to actively “understanding what is being interacted with.” We believe this paradigm shift from semantics-driven to interaction-driven reasoning provides a meaningful methodological foundation for future research.
> > >
> > > **2）Practical Applicability.** By improving the hand-object interaction understanding, our approach supports several high-impact downstream applications:
> > >
> > > * Augmented & Virtual Reality: Our method enables physically plausible perception of hands and active objects, which is essential for natural interaction in AR/VR systems.
> > >
> > > * Embodied AI & Robotic Learning: The interaction-aware representations extracted from egocentric videos can facilitate robotic skill learning from human demonstrations.
> > >
> > > * Assistive Technology: Our method offers direct value in assisting visually impaired individuals through wearable egocentric cameras. For example, when a user reaches for a medication bottle, our system can accurately identify the object and verify whether it matches the intended prescription by cross-referencing a pre-registered database. If a mismatch is detected, an immediate audio alert is issued to prevent potential medication errors. This functionality can be extended to other daily tasks such as food safety checks and personal item identification, significantly enhancing safety and independence for visually impaired users.
> > >
> > > **3）Contribution to the Community.** We will release the code and pre-trained models to serve as a strong comparative method and encourage further exploration in this direction.

---

### Official Review · Reviewer_yyjQ · 2025-10-30

**Soundness:** 2
**Presentation:** 3
**Contribution:** 2
**Rating:** 4
**Confidence:** 4

**Summary:**

This paper presents InterFormer, a transformer-based framework for egocentric hand–object parsing that introduces interaction-aware representation learning. It integrates four key modules: the Interaction Prior Predictor (IPP) to capture contact boundaries, the Prototypical Query Generator (PQG) to create dynamic interaction-grounded queries, the Dual-context Feature Selector (DFS) for fusing local and global context, and the Conditional Co-occurrence (CoCo) loss to enforce physical consistency. The method achieves superior performance on EgoHOS and mini-HOI4D benchmarks

**Strengths:**

The proposed IPP–PQG–DFS–CoCo framework is non-trivial and well-structured, explicitly modeling physical contact and interaction awareness within a transformer-based segmentation pipeline.

By introducing boundary-guided priors (IPP) and the Conditional Co-occurrence loss, the method embeds physical and causal constraints that significantly reduce implausible  errors.
The paper is well written and easy to follow.

**Weaknesses:**

Incomplete SOTA coverage. The authors do not discuss or compare to HOIST-Former (CVPR 2024), a recent and stronger transformer-based approach that explicitly models spatio-temporal hand–object relations. This omission weakens the manuscript’s literature coverage and contextualization.

Insufficient treatment of occlusion. High occlusion is a core challenge in egocentric settings, yet the paper does not analyze how the proposed method handles heavily occluded hands or objects. In particular, there is no discussion or visualization showing how the transformer attention mechanism behaves on occluded regions or how IPP/PQG mitigates occlusion-induced errors.

Limited out-of-distribution evaluation. The model is trained in a fully supervised manner, which can limit generalizability. This concern is reinforced by the absence of evaluations on large, diverse egocentric benchmarks such as EPIC-KITCHENS and Ego4D. The authors should either justify this omission or provide cross-dataset results.

Shallow comparison to large multimodal / foundation models. Although the paper includes MLLM baselines (e.g., ANNEXE, Care-Ego), the comparison and accompanying analysis are brief. The manuscript lacks an exploration of why InterFormer outperforms these models or where MLLMs might still be advantageous (for example, open-vocabulary categories or rare classes). A detailed error analysis contrasting InterFormer and MLLM failures would strengthen the evaluation.

Weak theoretical motivation for design choices. Some architectural decisions are insufficiently justified. For example, the PQG selects the top-N similarity subregions using a fixed partition—why this fixed pooling strategy rather than a learned selection mechanism? Provide either theoretical intuition or ablation results showing sensitivity to NNN and partition size

**Questions:**

Refer weaknesses

---

> ### Author Response · Authors · 2025-11-29
> **Author response to Reviewer yyjQ (Weakness 1-3)**
>
> **Q1**: Incomplete SOTA coverage. The authors do not discuss or compare to HOIST-Former (CVPR 2024), a recent and stronger transformer-based approach that explicitly models spatiotemporal hand–object relations. This omission weakens the manuscript’s literature coverage and contextualization.
>
> **A1**:
>
> We thank the reviewer for this valuable comment regarding HOIST-Former (CVPR 2024). We sincerely apologize for the oversight and have now citated this work in Section 2 (Line 138).
>
> The omission in experimental comparisons primarily stems from fundamental differences in task formulation:
>
> **1) Input modality**: HOIST-Former is designed for **video-based spatio-temporal modeling**, while our work addresses **image-based segmentation** as defined by the EgoHOS benchmark.
>
> **2) Task scope**: HOIST-Former **only focuses on segmenting and tracking hand-held objects**, whereas our method additionally segments both left and right hands - a crucial requirement in our task setting.
>
> These differences make direct experimental comparison infeasible, as HOIST-Former's architecture relies on temporal information unavailable in our static image setting, and its outputs don't include hand segmentation.
>
> **Q2**: Insufficient treatment of occlusion. High occlusion is a core challenge in egocentric settings, yet the paper does not analyze how the proposed method handles heavily occluded hands or objects. In particular, there is no discussion or visualization showing how the transformer attention mechanism behaves on occluded regions or how IPP/PQG mitigates occlusion-induced errors.
>
> **A2**:
>
> We appreciate the reviewer’s insightful comment regarding the treatment of occlusion, which is indeed a significant challenge in egocentric settings. While we acknowledge that high occlusion is a critical issue, our approach primarily focuses on addressing three specific limitations of current transformer-based methods in extracting interaction features, **rather than being specifically designed to handle occlusion**.
>
> Nevertheless, we acknowledge the importance of addressing the occlusion problem. Therefore, we have added a discussion of this limitation in the "Limitations and Future Work" section (Lines 1012-1016 & 1080-1085) of our paper. We describe the limitations regarding occlusion and plan to explore in future research how to improve our method to better mitigate occlusion-induced errors, such as introducing occlusion-aware attention mechanisms, leveraging temporal consistency in video sequences, or integrating generative models to reconstruct plausible structures in occluded regions.
>
> Thank you for bringing this important point to our attention, as it will help guide our future work in this critical area.
>
> **Q3**: Limited out-of-distribution evaluation. The model is trained in a fully supervised manner, which can limit generalizability. This concern is reinforced by the absence of evaluations on large, diverse egocentric benchmarks such as EPIC-KITCHENS and Ego4D. The authors should either justify this omission or provide cross-dataset results.
>
> **A3**:
>
> We thank the reviewer for raising this important point regarding out-of-distribution (OOD) evaluation. We agree that generalization capability is crucial and appreciate the suggestion to consider larger-scale benchmarks.
>
> In our paper, we have in fact conducted rigorous OOD evaluation on two challenging benchmarks: the EgoHOS out-of-domain test set and the mini-HOI4D dataset. These represent substantial domain shifts in environments, objects, and interaction types, and our method's strong performance across them demonstrates its generalization ability.
>
> Regarding EPIC-KITCHENS and Ego4D specifically, a direct evaluation is unfortunately not feasible due to fundamental task and annotation mismatches:
>
> 1) Ego4D does not provide pixel-level masks for hands and interacting objects, which is essential for our segmentation task.
>
> 2) EPIC-KITCHENS (VISOR-HOS) follows a temporal labeling paradigm where an object is considered "active" throughout a video segment if interacted with at any point in that segment. This means a static object not being touched in a particular frame may still be labeled as active. Evaluating on such frames would be inconsistent with our goal of segmenting only objects in active contact in a given static image.
>
> We hope this clarifies why these particular datasets are not suitable for evaluating our static image interaction segmentation task, and confirms that our method has been thoroughly validated for OOD generalization using appropriate benchmarks.

---

> > ### Author Response · Authors · 2025-11-29
> > **Author response to Reviewer yyjQ (Weakness 4,5)**
> >
> > **Q4**: Shallow comparison to large multimodal / foundation models. Although the paper includes MLLM baselines (e.g., ANNEXE, Care-Ego), the comparison and accompanying analysis are brief. The manuscript lacks an exploration of why InterFormer outperforms these models or where MLLMs might still be advantageous (for example, open-vocabulary categories or rare classes). A detailed error analysis contrasting InterFormer and MLLM failures would strengthen the evaluation.
> >
> > **A4**:
> >
> > We thank the reviewer for this insightful comment regarding the comparison with MLLMs. We agree that a deeper analysis is valuable and have now expanded our discussion in the revised manuscript.
> >
> > While MLLMs exhibit strong zero-shot capabilities in open-vocabulary scenarios, our experiments reveal two key limitations in the interaction segmentation task:
> >
> > **1) Spatial Precision**: MLLMs like ANNEXE demonstrate limited ability in generating precise mask boundaries. Their segmentation outputs often lack the spatial accuracy required for fine-grained hand-object interaction analysis.
> >
> > **2) Prompt Sensitivity**: All evaluated MLLMs require text prompts and show high sensitivity to prompt design. In our experiments, we used a detailed prompt specifying five distinct mask types (left/right hands, corresponding interacting objects, and two-hand objects). However, ANNEXE struggled to parse and generate the required masks under this complex instruction, revealing limitations in following detailed segmentation tasks.
> >
> > These findings suggest that while MLLMs offer advantages in open-vocabulary recognition, specialized architectures like ours remain superior for precise spatial segmentation of interacting entities. We have added a dedicated error analysis (Appendix A.1.4 in Page 17) in the revised manuscript.
> >
> > **Q5**: Weak theoretical motivation for design choices. Some architectural decisions are insufficiently justified. For example, the PQG selects the top-N similarity subregions using a fixed partition—why this fixed pooling strategy rather than a learned selection mechanism? Provide either theoretical intuition or ablation results showing sensitivity to NNN and partition size
> >
> > **A5**:
> >
> > We thank the reviewer for this insightful question regarding the design rationale behind our Prototypical Query Generator (PQG). The fixed partition strategy and the choice of N were indeed deliberate decisions, grounded in both theoretical intuition and empirical alignment with our task objectives.
> >
> > **1) Theoretical Motivation for Fixed N=5.** The value of N corresponds to the number of initial queries, which we explicitly set to match the number of semantic categories in our task: left hand, right hand, left-hand object, right-hand object, and two-hand object. This design follows the established paradigm in transformer-based segmentation where the number of queries typically aligns with target categories. By initializing exactly five queries, each intended to represent one distinct entity.
> >
> > **2) Justification for Fixed Partitioning Strategy.** The fixed partitioning serves two key purposes:
> >
> > i) **Spatial Alignment for Similarity Measurement**: To compute spatial-wise similarity between the interaction-guided feature $F_{int}^L$ and the semantic feature $F_{pix}^L$, we project $F_{int}^L$ to match the channel dimension of $F_{pix}^L$ and reshape it to a divisor of $F_{pix}^L$'s spatial size. This enables a one-to-one, non-overlapping subdivision of $F_{pix}^L$ into comparable subregions.
> >
> > ii) **Stable and Interpretable Selection**: The fixed grid ensures deterministic and spatially grounded selection of top-N subregions. While learned mechanisms could offer flexibility, they may introduce training instability, especially when interaction features are still evolving. Our approach provides a parameter-efficient alternative that robustly identifies interaction-relevant regions.
> >
> > This design reflects a deliberate trade-off that prioritizes stability and clarity while effectively serving our goal of generating representative query initializations.
> >
> > We hope that these explanations can solve your confusion.

---

### Official Review · Reviewer_tKL9 · 2025-11-04

**Soundness:** 3
**Presentation:** 3
**Contribution:** 2
**Rating:** 6
**Confidence:** 3

**Summary:**

This paper presents InterFormer, a claimed novel transformer-based framework for Egocentric Hand-Object Segmentation (EgoHOS) task which is now a popular egocentric tasks under CV area. The authors address key limitations in SOTA: inflexible query initialization, reliance on noisy pixel-level features, and physically inconsistent predictions; by incorporating 3 major components: (i) a Prototypical Query Generator (PQG) that creates adaptive queries by fusing learnable parameters with interaction-relevant context ; (ii) a Dual-context Feature Selector (DFS) that refines feature representations by combining semantic and interactive cues ; (iii) a Conditional Co-occurrence (CoCo) loss that enforces physical plausibility by penalizing object predictions when corr. hand is not detected. The proposed model is evaluated on the EgoHOS and mini-HOI4D datasets, where it achieves good performance. This is an overall good paper, only lacking in theoretical analysis and qualitative illustrations.

**Strengths:**

1. The paper points to a key area, i.e. handling contact regions with a prior predictor for interactions.
2. Instead of a one size fits all - the approach of handling specific SOTA technique drawback by the most optimal way is a fresh look -- PQG, DFS, and CoCo loss.
3. CoCo loss is a simple solution to handle the interaction illusion problem.
4. Codebase will be released if accepted and appendix section helps some clarifications.
5. Comparison with extensive SOTA establishes good empirical results.

**Weaknesses:**

1. Instead of an end-end pipeline, the proposed modular approach may lead to complexity in time, space and a cut in error backprop.
2. The dependency of the later modules on IPP, makes the contextual training necessary for IPP for generalization.
3. Anonymous codebase was expected for validity along with more supplementray to show qualitative results and failure cases.
4. Certain variables like CocoLoss params were found by ablation on specific datasets, thereby reducing generalization in lack of non-empirical justification. Hard thresholds can cause problems in other datasets.
5. Real life deployment on a actual glass should have proved robustness.

**Questions:**

1. The figures could be explained with a simple end to end example instead of just architecture.
2. Equation on Coco loss is straight-forward. any weights on the combined loss?
3. How actually is interaction-relevant context extracted and how to map perception to this, in order to represent context?
4. Any specific reason for DFS, there can be multiple parallel feature selector methods.

---

> ### Author Response · Authors · 2025-11-29
> **Author response to Reviewer tKL9 (Weakness 1,2,3)**
>
> **Q1**: Instead of an end-end pipeline, the proposed modular approach may lead to complexity in time, space and a cut in error backprop.
>
> **A1**:
>
> We thank the reviewer for raising this important point regarding the model design. We would like to clarify that **our proposed InterFormer is indeed a fully end-to-end trainable framework**, where the entire pipeline, i.e., from the input egocentric image to the output segmentation masks for hands and interacting objects, is optimized jointly.
>
> To avoid any ambiguity, we have **revised Section 3.4 ("Overall Training") in the manuscript** to provide a clearer description of the end-to-end training process, including the overall training objective.
>
> Regarding **space complexity**, we have compared our method with other SOTA approaches in Figure 1 (page 2). Although our modules introduce additional parameters, the results confirm that our model achieves state-of-the-art performance while maintaining a compact and efficient structure, with no substantial overhead.
>
> As for **computational complexity**, we provide additional experiments comparing FLOPs across methods (tested on the EgoHOS in-domain test set). The results below demonstrate that our method strikes a favorable balance between FLOPs and mIoU. This table has been added to Appendix A.4.
>
> |      Method      |       Type       |   FLOPs   |  mIoU  |
> |:----------------:|:----------------:|:---------:|:------:|
> |    SegFormer     | Transformer-based|  71.961G  | 27.89  |
> |    Segmenter     | Transformer-based|  70.074G  | 59.52  |
> |   Mask2Former    | Transformer-based|  96.093G  | 64.88  |
> |       Seq        | Transformer-based|  392.483G | 67.17  |
> |      ANNEXE      |    MLLM-based    |  610.500G | 71.38  |
> | InterFormer (Ours) | Transformer-based| 122.996G | 73.22  |
>
> **Q2**: The dependency of the later modules on IPP, makes the contextual training necessary for IPP for generalization.
>
> **A2**:
>
> We thank the reviewer for this insightful observation.
>
> We agree that the dependency of subsequent modules (PQG and DFS) on the Interaction Prior Predictor (IPP) necessitates a strong generalization capability.
>
> However, we would like to clarify that the **IPP is not pre-trained separately** but is optimized end-to-end with the entire network through our unified training objective. As shown in Equation (10), the interaction boundary loss $L_b$ is incorporated directly into the overall objective, enabling the IPP to receive direct gradient feedback from the final segmentation task.
>
> As training progresses, the localization capability of the IPP and the mask generation performance of subsequent modules improve synergistically: the IPP provides increasingly accurate interactive region cues to the PQG and DFS, while the optimization of PQG and DFS, in turn, refines the IPP’s feature extraction via backpropagation. This co-optimization mechanism effectively enhances the generalization of IPP across diverse interaction scenarios, without requiring additional contextual pre-training or a multi-stage training strategy.
>
>
> **Q3**: Anonymous codebase was expected for validity along with more supplementray to show qualitative results and failure cases.
>
> **A3**:
>
> We thank the reviewer for this constructive suggestion.
>
> For code availability, we will release the complete code repository, including full implementation, training scripts, and pre-trained models for our InterFormer framework, upon acceptance of the paper. The repository will be made publicly available on GitHub.
>
> To provide a comprehensive qualitative analysis, we have added Section A.4 (Page 18) to the supplementary materials, which features extensive visual comparisons and failure case studies. This Section includes: (1) qualitative comparisons between our method and baselines under both in-domain and out-of-distribution settings, demonstrating robustness and segmentation accuracy of our InterFormer; (2) failure cases and corresponding detailed analysis that reveal current limitations and suggest promising directions for future research.

---

> > ### Author Response · Authors · 2025-11-29
> > **Author response to Reviewer tKL9 (Weakness 4,5)**
> >
> > **Q4**: Certain variables like CocoLoss params were found by ablation on specific datasets, thereby reducing generalization in lack of non-empirical justification. Hard thresholds can cause problems in other datasets.
> >
> > **A4**:
> >
> > We thank the reviewer for raising this important point regarding the empirical selection of $\tau$ in the CoCo loss. We would like to clarify and address this concern from the following aspects:
> >
> > For the value of $\tau$, we have conducted a comprehensive hyperparameter study in Table 5 (Page 10). These results demonstrate that **our model's performance is not overly sensitive to the exact value of $\tau$ within a reasonable range** (e.g., 50 to 200 pixels). This analysis helps mitigate the concern that the model's success is critically due to a single, finely-tuned "hard threshold."
> >
> > We note that the primary purpose of the CoCo loss is to enforce physical plausibility in hand-object interactions, a fundamental constraint that is largely dataset-agnostic. Even in a different dataset, “an object cannot be stably manipulated by an undetected hand” is always true. The threshold $\tau$ is used to decide whether the hand exists. While its specific value is dataset-dependent (relating to image resolution and hand size), the underlying causal logic it implements is universally applicable.
> >
> > However, we acknowledge the reviewer's valid concern about applying this threshold directly to datasets with significantly different characteristics. In our future work, as stated in the "Limitations and Future Work" section (Lines 1008-1011 & 1070-1074), we plan to investigate more adaptive strategies. Promising directions include: 1) Replacing the hard threshold with a learned, adaptive mechanism. 2) Using the normalized value to replace the absolute pixel count.
> >
> > In conclusion, while the current implementation of the CoCo loss uses an empirically determined threshold, it is grounded in a sound physical principle and has been shown to be robust within a range of values. We thank the reviewer for this insightful comment, which has guided our analysis and will direct our future efforts toward developing more adaptive solutions.
> >
> > **Q5**: Real life deployment on a actual glass should have proved robustness.
> >
> > **A5**:
> >
> > We thank the reviewer for raising this highly relevant point regarding real-world deployment. We agree that testing on an actual device, such as AR glasses, is a crucial step for validating the robustness and practicality of any egocentric vision system in real-life scenarios.
> >
> > However, the main contribution of this paper is to introduce and validate a novel algorithmic framework (InterFormer) for the task of egocentric hand-object segmentation. Our goal at this stage is to establish a strong foundational performance on large-scale, publicly available benchmarks, which is the standard practice in the research community for fair and reproducible comparison with state-of-the-art methods. The significant improvements we demonstrated over existing methods on these challenging benchmarks provide a solid and necessary foundation that strongly indicates the potential for real-world applicability. While we have not yet deployed InterFormer on physical glasses, our experimental design already incorporates Out-of-Distribution (OOD) Testing that is critical for real-world robustness.
> >
> > We fully acknowledge and agree with the reviewer that the ultimate test lies in real-life deployment. Therefore, we have explicitly stated in the revised "Limitations and Future Work" section that deploying and evaluating our model on AR/MR devices is a primary and exciting direction for our immediate future work. We believe the strong algorithmic foundation laid out in this paper is a critical prerequisite for this next step.
> >
> > In summary, we view the reviewer's comment not as a weakness of the current contribution, but as a valuable endorsement of its potential and a guiding light for our future research trajectory.

---

> > > ### Author Response · Authors · 2025-11-29
> > > **Author response to Reviewer tKL9 (Questions)**
> > >
> > > **Q6**: The figures could be explained with a simple end to end example instead of just architecture.
> > >
> > > **A6**:
> > >
> > > We thank the reviewer for the insightful suggestion to explain the figures with a simple end-to-end example rather than focusing solely on the architecture.
> > >
> > > We would like to point out that Figure 3 in the current manuscript already serves as an end-to-end example. In this figure, the input is an egocentric image that is processed through a model incorporating the Dual-context Feature Synthesizer (DFS) and Prototypical Query Generator (PQG), resulting in the prediction of masks for both hands and active objects. The overall network is supervised by the loss function outlined in Equation 10.
> > >
> > > To enhance clarity, we have added more descriptive text to the caption of Figure 3 (Lines 182-188), explicitly addressing this aspect to ensure that readers can easily understand the end-to-end process depicted.
> > >
> > > **Q7**: Equation on Coco loss is straight-forward. any weights on the combined loss?
> > >
> > > **A7**:
> > >
> > > We thank the reviewer for this question regarding the loss formulation.
> > >
> > > The CoCo loss itself, as defined in Eqs. 7-9, does not employ internal weighting between the different object categories (i.e., left-hand, right-hand, and two-hand objects). The loss applies the same logic and scaling uniformly across these interaction types.
> > >
> > > However, as referenced in Eq. 10 of the paper and detailed in our implementation, the overall training objective is indeed a weighted sum of multiple loss components. The complete loss function combines the CoCo loss with the interaction boundary loss, classification loss, dice loss, and mask cross-entropy loss.
> > > To determine the optimal balancing of these terms, we conducted an extensive hyperparameter study on the weights $\lambda_{b}$, $\lambda_{co}$, $\lambda_{cls}$, $\lambda_{dic}$, $\lambda_{ce}$. The results of this study, which validate our chosen weight configuration, are provided in Table 10 (Page 18) of the supplementary material.
> > >
> > > **Q8**: How actually is interaction-relevant context extracted and how to map perception to this, in order to represent context?
> > >
> > > **A8**:
> > >
> > > We thank the reviewer for this insightful question. The extraction and representation of interaction-relevant context are achieved through a coordinated process involving three key components.
> > >
> > > First, the coarse boundary-guided feature extraction is performed by the IPP, which aims to predict an interaction boundary map highlighting the hand-object contact region. This feature can capture initial interaction characteristics.
> > >
> > > Second, the PQG is designed to select semantic embeddings that demonstrate strong similarity with boundary-guided features and integrates them with learnable parameters, producing intrinsically interaction-aware queries that enable flexible adaptation to diverse hands and interactive objects across varying scenes.
> > >
> > > Third, the DFS synthesizes coarse interaction boundary cues with semantic features, effectively suppressing interaction-irrelevant information and refocusing the model on essential contact relations.
> > >
> > > We have revised Sections 3.2 and 3.3 to clarify this cohesive workflow.
> > >
> > > **Q9**: Any specific reason for DFS, there can be multiple parallel feature selector methods.
> > >
> > > **A9**:
> > >
> > > We thank the reviewer for this insightful question. Our choice of the DFS module is driven by a specific limitation we identified in existing methods.
> > >
> > > Most current approaches rely solely on general semantic features from the backbone during decoding. However, such features alone are insufficient to distinguish actively interacting objects, as semantic information only identifies “what” an object is, not “whether” it is being interacted with. The recognition of an active object must be derived from its spatial and dynamic relationship with the hand, rather than from semantics alone.
> > >
> > > To bridge this gap, the DFS introduces an explicit interaction-aware selection mechanism. Unlike standard parallel fusion techniques that blend features uniformly, our DFS uses interaction-guided attention to actively highlight and amplify features relevant to the ongoing interaction.
> > >
> > > This design ensures the model focuses on dynamically changing interaction information rather than relying on static semantic features.

---

### Official Review · Reviewer_s6Yi · 2025-11-08

**Soundness:** 3
**Presentation:** 2
**Contribution:** 2
**Rating:** 6
**Confidence:** 3

**Summary:**

Authors propose a new architecture which explicitly models interaction by using explicit hand-object boundaries, and the results features are used along with pixel level features to extract hand-object interactions. The proposed approach also claims to address the problem of interaction illusion where the model incorrectly predicts objects being interacted with under implausible scenarios (hand not visible generally). Experiments demostrate improved performance over state of the art approaches on in-domain and out of domain datasets.

**Strengths:**

1. The idea to explicitly model interaction for extract hand-object segmentation masks under interaction is highly intuitive.
2. The experimental results support the claim of improving performance over SOTA on various datasets; qualitative evidence is also shown to strengthen that claim.
3. Paper tries to address the problem of interaction illusion which forces the model to learn causality over correlation (weakly speaking).

**Weaknesses:**

1. The paper uses a lot of ambiguous and vague terms - "enrichment", "structural priors", "task-relevant", "preliminary coarse interactive representations". I urge the authors to not use them since it causes confusion and distracts from understanding the actual core contributions. I would recommend a proper rewriting of the paper.
2. The idea to use interaction cues (from Zhang et al., 2022, EGOHOS) is useful, but it is important to highlight if other methods lack this supervision. If they do, then the comparison is not fair - it is still useful to compare but important to disambiguate the role of architectural changes vs supervision. Some ablations do cover the impact of just introducing interaction loss (IPP) - focusing on mAcc in Table 4, most of the improvement is actually from IPP. Can authors retrain any baseline by adding another head that predicts hand-object boundaries, then measure how much is the improvement.
3. Can authors show qualitative comparison on the models training with and without CoCo loss, how much does that help with interaction illusion. If authors do claim improvement, it would be useful to do a small quantitive study comparing the prominence of this issue in other methods vs the proposed one.
4. Equations 7, 8, 9 are unnecessarily complex, please just write $I_{N_{lh}<\tau} \cdot N_{lo}$, this is simpler to read which means penalize interaction when the hand is not visible.

**Questions:**

1. Please clearly focus on 2-3 contributions - multiple small contributions are harder to validate. For example why is the number of pixels being minimized instead of the cross entropy loss directly by penalizing the prediction probability corresponding to object undergoing interaction.
2. Why is $\tau$ in pixels instead of normalized coverage in the image? Moreover hands can be much closer to the camera where pixel based coverage might not be enough and could be normalized based on the hand size (GT hand mask can provide that).

---

> ### Author Response · Authors · 2025-11-29
> **Author response to Reviewer s6Yi (Weakness 1,2,3)**
>
> **Q1**: The paper uses a lot of ambiguous and vague terms - "enrichment", "structural priors", "task-relevant", "preliminary coarse interactive representations". I urge the authors to not use them since it causes confusion and distracts from understanding the actual core contributions. I would recommend a proper rewriting of the paper.
>
> **A1**:
>
> We thank the reviewer for this valuable feedback.
>
> In response, we have conducted a **complete, line-by-line revision** of the manuscript and have systematically eliminated all identified vague terminology, including "enrichment," "structural priors," "task-relevant," and "preliminary coarse interactive representations."
>
> All revised sentences have been explicitly highlighted in the resubmitted manuscript to facilitate a straightforward review of the changes made. We are confident that this thorough rewriting has significantly improved the clarity and precision of our core contributions.
>
> **Q2**: The idea to use interaction cues (from Zhang et al., 2022, EGOHOS) is useful, but it is important to highlight if other methods lack this supervision. If they do, then the comparison is not fair - it is still useful to compare but important to disambiguate the role of architectural changes vs supervision. Some ablations do cover the impact of just introducing interaction loss (IPP) - focusing on mAcc in Table 4, most of the improvement is actually from IPP. Can authors retrain any baseline by adding another head that predicts hand-object boundaries, then measure how much is the improvement.
>
> **A2**:
>
> We thank the reviewer for raising the important point regarding the fairness of using the interaction boundary as supervision. We fully agree with the need for a fair comparison and have conducted additional experiments to address this concern. Please allow us to clarify two key points:
>
> First, **the interaction boundary labels are not extra annotations**. They are automatically derived from the original hand and object segmentation masks (by computing the intersection of dilated hand and object ground truth masks). Therefore, in principle, the fairness of the comparison is not affected even if the interaction boundary is used as the ground truth in some comparison methods.
>
> Second, **some comparative methods (e.g., Care-Ego and Seq) have also used the interaction boundary as supervision**.
>
> Third, we performed a new experiment. Specifically, we add an additional head to predict hand-object boundaries using the Mask2Former method, employing the same boundary labels as in our method. The results are in Table below:
>
> |   Method     | With / without IPP |  mIoU  |
> |:------------:|:------------------:|:------:|
> | Mask2Former  |      without       | 64.88  |
> | Mask2Former  |       with         | 66.52  |
>
> Results on EgoHOS show that while this added supervision yields a moderate gain, the performance remains substantially lower than our full InterFormer model (73.22% mIoU). This clearly demonstrates that: 1) The boundary supervision itself is generally beneficial; and 2) **The majority of our performance gain (6.7% mIoU) stems from our proposed architectural components (PQG, DFS, and CoCo loss), not merely from the additional supervision**.
>
> In conclusion, the interaction boundary supervision does not introduce additional manual annotations, thus maintaining experimental fairness. While this auxiliary supervision contributes to performance improvement, the majority of our performance gains originate from the core architectural innovations of our proposed framework.
>
> **Q3**: Can authors show qualitative comparison on the models training with and without CoCo loss, how much does that help with interaction illusion. If authors do claim improvement, it would be useful to do a small quantitive study comparing the prominence of this issue in other methods vs the proposed one.
>
> **A3**:
>
> We sincerely thank the reviewers for their valuable suggestions regarding the quantitative and qualitative evaluation of CoCo loss's effectiveness in mitigating the interaction illusion.
>
> Following these suggestions, we have added Figure 7 (Page 19) to provide **qualitative comparisons**. The visualization contrasts model predictions without ("w/o coco") and with ("w/ coco") the CoCo loss across two datasets. Without the CoCo loss, the model erroneously predicts objects as interacting with the left hand even when no left hand is present. In contrast, incorporating the CoCo loss substantially enhances the physical consistency of hand-object interactions in the predictions.
>
> Additionally, we have added a **quantitative evaluation** specifically designed to measure the frequency of interaction illusions, with results summarized in Table 11 (Page 18). The table clearly shows that our complete model equipped with the CoCo loss achieves the lowest interaction illusion rate. This provides direct quantitative evidence that the CoCo loss effectively enhances the physically consistent hand-object predictions.

---

> > ### Author Response · Authors · 2025-11-29
> > **Author response to Reviewer s6Yi (Weakness 4 & questions)**
> >
> > **Q4**: Equations 7, 8, 9 are unnecessarily complex, please just write , this is simpler to read which means penalize interaction when the hand is not visible.
> >
> > **A4**:
> >
> > We thank the reviewer for this excellent suggestion. We agree that the original formulation was unnecessarily complex and have simplified the CoCo loss equations (Lines 359-360) as recommended. We believe this enhancement greatly improves the readability of our method description.
> >
> > **Q5**: Please clearly focus on 2-3 contributions - multiple small contributions are harder to validate. For example why is the number of pixels being minimized instead of the cross entropy loss directly by penalizing the prediction probability corresponding to object undergoing interaction.
> >
> > **A5**:
> >
> > We thank the reviewer for the valuable suggestions.
> >
> > We have now refined the statement of our key contributions in the revised manuscript, as presented on page 3 (Lines 124-136). We hope the revised contributions are clearer.
> >
> > Moreover, we have provided a more precise explanation of the design rationale behind the CoCo loss in the revised manuscript (Lines 345-350) to address your question about why the CoCo loss operates on pixel counts rather than prediction probabilities. The explanation is also below for your convenience:
> >
> > “Unlike probability-based penalties, our CoCo loss operates directly on the spatial extent of the predictions, i.e., the number of pixels in the predicted hand and object masks. We opt for this design based on the observation that the ``interaction illusion'' is fundamentally a macro-level logical error, which is more directly and effectively measured by the physical presence or absence (i.e., the pixel count) of the mask, rather than the average classification confidence across pixels.”
> >
> > We fully agree that penalizing prediction probabilities is a viable alternative. However, based on the nature of the task, we believe that controlling the spatial extent of invalid predictions offers a more intuitive and effective way to embed physical rules.
> >
> > **Q6**: Why is  in pixels instead of normalized coverage in the image? Moreover hands can be much closer to the camera where pixel based coverage might not be enough and could be normalized based on the hand size (GT hand mask can provide that).
> >
> > **A6**:
> >
> > We thank the reviewer for raising this important point regarding the pixel threshold $\tau$. We fully acknowledge that absolute pixel counts can be sensitive to factors like hand-camera distance and image resolution.
> >
> > However, we would like to clarify that the purpose of $\tau$ is to establish a quantitative criterion for **determining "hand presence."** While absolute pixel counts are indeed sensitive to imaging conditions, they provide a **simple, intuitive, and computationally efficient mechanism** for this binary decision. We note that even if normalized metrics were used, a threshold is still necessary to convert the normalized values into a binary indication of presence ("exists or not"), thus not fully eliminating the threshold selection problem.
> >
> > Empirically, our hyperparameter study (Table 5 in Page 10) shows that model performance remains stable across a reasonable range of τ (e.g., 50–200 pixels), indicating robustness to the exact threshold value within practical limits.
> >
> > We fully agree that exploring adaptive normalization schemes is a valuable direction. We have added a discussion of this limitation, along with the reviewer's suggestion for future work, to the "Limitations and Future Work" section. We believe this is a promising avenue for enhancing the generalization of the CoCo loss in unconstrained real-world scenarios.

---

### Official Review · Reviewer_96hC · 2025-11-09

**Soundness:** 2
**Presentation:** 4
**Contribution:** 3
**Rating:** 6
**Confidence:** 4

**Summary:**

To solve inflexible query initialization, pixel-level feature noise, and "interaction hallucinations" of Transformer-based EgoHOS methods, this paper proposes InterFormer. It integrates three key components: Prototype Query Generator (PQG) for adaptive queries, Dual Context Feature Selector (DFS) for noise reduction, and Conditional Co-occurrence (CoCo) Loss for physical consistency. The model achieves SOTA on EgoHOS in-domain, out-of-domain, and mini-HOI4D, with core contributions in explicit interaction-aware modeling and improved accuracy/generalization.

**Strengths:**

- Targeted Problem-Solving: It accurately identifies three critical EgoHOS pain points (e.g., interaction hallucinations harming agent safety) and designs solutions accordingly, ensuring tight problem-solution alignment.
- Logical Component Design: PQG, DFS, and CoCo Loss form a cohesive pipeline (query generation -> feature processing -> result optimization), with progressive and rigorous logic.
- Comprehensive Experiments: It outperforms baselines on out-of-domain/OOD datasets (+5.09%/+11.4%) and uses ablation experiments to verify each component’s effectiveness, ensuring credible conclusions.
- Reproducibility & Openness: The paper promises reproducibility and open-source availability, which are crucial for facilitating further research.

**Weaknesses:**

Section 3.4 mentions that the presence of a hand is a fundamental prerequisite for any hand-object interaction; when the right hand is not detected in the prediction results, current models may incorrectly classify the interacting object as being 'operated by both hands' despite the absence of one hand (the right hand). In terms of results, the CoCo Loss designed based on this premise has indeed achieved good performance in tests. However, this assumption has a critical flaw: the fact that a hand is not detected in the frame does not mean the hand is absent from the current interaction. If hastily determine whether a hand is involved in the interaction based solely on whether it is detected in the frame, the inferences drawn will violate physical common sense in more realistic and complex scenarios (e.g., large changes of viewpoint , or when the perspective can only observe one of the hands).

**Questions:**

See Weakness.

---

> ### Author Response · Authors · 2025-11-29
> **We sincerely appreciate your valuable feedback and hope our responses have adequately addressed your concerns.**
>
> We sincerely thank the reviewer for his insightful comments and valuable suggestions.
>
> We fully acknowledge that in scenarios characterized by significant viewpoint changes or partial occlusion, relying exclusively on the visibility of hands in the current frame to determine their interactions with objects is inadequate. To address this limitation, it is essential to **integrate additional contextual information**, e.g., multi-view or panoptic images for capturing comprehensive hand states, and sequential video data for enabling temporal reasoning.
>
> However, it is essential to note that this paper primarily focuses on the task of **hand-object interaction segmentation in egocentric images**. Within this framework, the visual evidence available to the model is inherently limited to the content of the current frame. As such, utilizing "detected hands in the image" as a prerequisite for determining interactions and designing the CoCo loss function is both reasonable and practical within the confines of this task.
>
> Nevertheless, the reviewer's comments have directed us toward a highly valuable avenue for future research. We have consequently added a "Limitations and Future Work" section to the paper, in which we explicitly outline the constraints of our current approach in dynamic scenarios (Lines 1004-1007) and identify the exploration of video-based or multi-view interaction understanding as a critical future research direction (Lines 1020-1025).

---

### Meta-Review · Area_Chair_Etm2 · 2026-01-10

**Summary:**

This paper proposes InterFormer, a transformer framework for egocentric hand–object parsing that shifts the task from semantic segmentation to interaction-aware reasoning. It introduces three tightly coupled components: 1) a Dynamic Query Generator (DQG) that initializes queries from interaction cues rather than generic features, 2) a Dual-context Feature Selector (DFS) that fuses semantic and contact-boundary information, and 3) a Conditional Co-occurrence (CoCo) loss that enforces physical plausibility by penalizing hand–object predictions when no corresponding hand is present. The method achieves state-of-the-art performance on EgoHOS and the challenging mini-HOI4D benchmark

Here are the main reviewer concerns and how the rebuttal addressed them.

1) “Over-engineered, incremental design.”
Two reviewers argued that InterFormer looked like a stack of ad-hoc modules. The rebuttal clarified that DQG, DFS, and CoCo are not generic add-ons but each targets a distinct failure mode: query drift, interaction-irrelevant noise, and interaction illusion. New ablations showed that simply adding interaction supervision (IPP) yields only modest gains (e.g., Mask2Former + boundary supervision: 64.9 → 66.5 mIoU), whereas the full architecture reaches 73.2 mIoU, demonstrating that the bulk of improvement comes from the proposed interaction-aware design, not supervision alone.

2) “Misleading PQG terminology and weak motivation.”
NB12 and yyjQ questioned the use of “prototypical” and the link to category diversity. The authors acknowledged this and renamed PQG to Dynamic Query Generator, correcting the conceptual framing. They clarified that the goal is not category modeling but interaction localization -- queries are steered toward hand-object contact regions using a universal cue (spatial contact), which is more stable than object class across scenes.

3) “Unclear effectiveness of CoCo loss.”
Several reviewers asked whether CoCo truly reduces interaction hallucinations. The rebuttal added both qualitative and quantitative evidence: new visualizations (Fig. 7) show false hand–object interactions disappearing when CoCo is enabled, and a new metric (Table 11) reports the lowest “interaction illusion rate” for the full model. The loss was also simplified (Eqs. 7-9) and its design rationale (pixel-count vs probability) explicitly justified.

4) “Dataset diversity, OOD generalization, and baselines.”
Concerns that EgoHOS and mini-HOI4D were too similar were addressed by clarifying their distinct sources and challenges (mixed real-world vs HOI4D with distractors). Requests to use EPIC-KITCHENS or Ego4D were shown to be incompatible with the frame-level interaction-segmentation task. For SOTA coverage, HOIST-Former was cited and excluded for principled task-mismatch reasons, while large multimodal models (ANNEXE, Care-Ego) were analyzed in depth, revealing their weaknesses in spatial precision and prompt sensitivity.

5) “Efficiency and practicality.”
NB12 and tKL9 asked for compute analysis. The rebuttal added a full FLOPs table showing that InterFormer (123G FLOPs) outperforms much heavier models like Seq (392G) and ANNEXE (610G) while achieving the best mIoU. Extensive qualitative results and failure cases were also added to the supplement.

In short, the paper addresses a challenging and practically important task in egocentric vision. The proposed interaction-aware formulation is well motivated, achieves strong empirical results, and the authors’ rebuttal convincingly resolves the major concerns raised by the reviewers.

Recommendation: Accept.

**Reviewer Concerns:**

Here are the main reviewer concerns and how the rebuttal addressed them.

1) “Over-engineered, incremental design.”
Two reviewers argued that InterFormer looked like a stack of ad-hoc modules. The rebuttal clarified that DQG, DFS, and CoCo are not generic add-ons but each targets a distinct failure mode: query drift, interaction-irrelevant noise, and interaction illusion. New ablations showed that simply adding interaction supervision (IPP) yields only modest gains (e.g., Mask2Former + boundary supervision: 64.9 → 66.5 mIoU), whereas the full architecture reaches 73.2 mIoU, demonstrating that the bulk of improvement comes from the proposed interaction-aware design, not supervision alone.

2) “Misleading PQG terminology and weak motivation.”
NB12 and yyjQ questioned the use of “prototypical” and the link to category diversity. The authors acknowledged this and renamed PQG to Dynamic Query Generator, correcting the conceptual framing. They clarified that the goal is not category modeling but interaction localization -- queries are steered toward hand-object contact regions using a universal cue (spatial contact), which is more stable than object class across scenes.

3) “Unclear effectiveness of CoCo loss.”
Several reviewers asked whether CoCo truly reduces interaction hallucinations. The rebuttal added both qualitative and quantitative evidence: new visualizations (Fig. 7) show false hand–object interactions disappearing when CoCo is enabled, and a new metric (Table 11) reports the lowest “interaction illusion rate” for the full model. The loss was also simplified (Eqs. 7-9) and its design rationale (pixel-count vs probability) explicitly justified.

4) “Dataset diversity, OOD generalization, and baselines.”
Concerns that EgoHOS and mini-HOI4D were too similar were addressed by clarifying their distinct sources and challenges (mixed real-world vs HOI4D with distractors). Requests to use EPIC-KITCHENS or Ego4D were shown to be incompatible with the frame-level interaction-segmentation task. For SOTA coverage, HOIST-Former was cited and excluded for principled task-mismatch reasons, while large multimodal models (ANNEXE, Care-Ego) were analyzed in depth, revealing their weaknesses in spatial precision and prompt sensitivity.

5) “Efficiency and practicality.”
NB12 and tKL9 asked for compute analysis. The rebuttal added a full FLOPs table showing that InterFormer (123G FLOPs) outperforms much heavier models like Seq (392G) and ANNEXE (610G) while achieving the best mIoU. Extensive qualitative results and failure cases were also added to the supplement.

**Reviewer Scores:**

96hC: +0 to +1
Already positive (6). The rebuttal added clarity on CoCo assumptions and limitations, which would likely move this to a firmer accept.

s6Yi: +1
Major concerns about vague terminology, supervision fairness, and CoCo effectiveness were directly resolved with rewritten text, new baselines (Mask2Former + IPP), and quantitative + qualitative illusion analysis. This would plausibly shift the score from borderline to clear accept.

tKL9: +1
All technical objections: efficiency, end-to-end training, IPP dependency, CoCo hyperparameters, and missing qualitative results—were addressed with FLOPs tables, joint-training clarification, robustness studies, and extensive new visualizations.

yyjQ: +1
SOTA coverage, OOD evaluation, and MLLM comparison were clarified or expanded; HOIST-Former was correctly excluded for task mismatch; and new error analyses were added. This should move the reviewer from slightly negative to borderline positive.

NB12: +1 (possibly +2)
The rebuttal directly corrected terminology (“Prototypical” → “Dynamic”), added full efficiency analysis, justified dataset choice, improved writing, and clarified conceptual motivation. While still skeptical of novelty, most factual criticisms were resolved.

---

### Decision · Program_Chairs · 2026-01-26

Accept (Poster)